

# Comparison between the assimilation of IASI Level 2 retrievals and Level 1 radiances for ozone reanalyses

Emanuele Emili[1], Brice Barret[2], Eric Le Flochmoën[2], and Daniel Cariolle[1]

[1]CECI, Université de Toulouse, Cerfacs, CNRS, Toulouse, France
[2]Laboratoire d'Aérologie, Université de Toulouse, CNRS, UPS, Toulouse, France

**Correspondence:** E. Emili (emili@cerfacs.fr)

**Abstract.** The prior information used for Level 2 (L2) retrievals in the thermal infrared can influence the quality of the retrievals themselves and, therefore, their further assimilation in atmospheric composition models. In this study we evaluate the differences between assimilating L2 ozone profiles and Level 1 (L1) radiances from the Infrared Atmospheric Sounding Interferometer (IASI). We minimized potential differences between the two approaches by employing the same radiative transfer

code (RTTOV) and a very similar setup for both the L2 retrievals (1D-Var) and the L1 assimilation (3D-Var). We computed hourly 3D-Var reanalyses assimilating respectively L1 and L2 data in the chemical transport model MOCAGE and compared the resulting $O_3$ fields among each other and against ozonesondes. We also evaluated the joint assimilation of limb measurements from the Microwave Limb Sounder (MLS) on top of IASI to assess the impact of stratospheric $O_3$ on tropospheric reanalyses. Results indicate that significant differences can arise between L2 and L1 assimilation, especially in regions where

the L2 prior is biased (at the tropics and in the southern hemisphere in this study). L1 and L2 assimilation give instead very similar results in the northern hemisphere, especially when MLS measurements are used to constrain the stratospheric $O_3$ column. We conclude this study by listing remaining issues that are common to both the L1 and L2 approaches and that deserve further research.

## 1 Introduction

The global monitoring of the atmospheric composition relies on a large number of dedicated satellite missions and on the sustained improvement of numerical forecast models. Research and operational centers provide today both satellite based reanalyses and forecasts of atmospheric composition for a large number of applications, spanning from stratospheric ozone monitoring (van der A et al., 2010) to climate change (Flemming et al., 2017) and air-quality (Zhang et al., 2012; Marécal et al., 2015).

Satellite sensors measure the spectral signature of gases and aerosols on the radiation field that traverse the atmosphere. Retrieving the concentration of a given gas from the radiation measured at the satellite position represents an inverse problem that is in most cases ill-posed and under-determined, i.e. finding the solution requires some type of mathematical regularization or prior information (Rodgers, 2000). The accuracy of the solution depends in general on the intensity of the spectral signature of the retrieved compound, the source of radiation (e.g. the Earth or the Sun), the observation geometry and the accuracy of the



Radiative Transfer Modeling (RTM). The latter means also correctly accounting for all the atmospheric constituents or surface properties that affect the radiation field but are not retrieved themselves (auxiliary RTM inputs).

When the retrieval is done within a Bayesian framework, like the optimal estimation method (Rodgers, 2000), the measurements errors, the RTM errors and the uncertainty in the prior information (also named background or a-priori) are prescribed. The procedure provides then an estimation of the error covariance for the retrieved quantity and the Degrees Of Freedom (DOF) of the solution. The retrieval errors and DOF can be used first to diagnose the quality and the relevance of the atmospheric retrieval. They become even more important when retrievals are further assimilated in numerical forecast models, because they weight the impact of the observations in the system.

First atmospheric composition models were named Chemical Transport Models (CTM) because they solve the chemical and physical processes but are based on meteorological fields computed separately with a Numerical Weather Prediction (NWP) model. Coupled Chemistry Meteorology Models (CCMM) that simulate both meteorology and chemistry online became available later but are today quite common in operational centers (Zhang, 2008; Flemming et al., 2015). There are currently growing efforts to introduce even stronger coupling of the atmosphere with both ocean and surface models, which gives so-called Earth System Models (ESM) (Brown et al., 2012; Hurrell et al., 2013). ESMs provide a comprehensive tool for climate predictions and reanalyses, but they are also considered for state-of-the-art air quality modeling (Neal et al., 2017).

Following closely the historical advances in modeling, the assimilation of satellite data has been introduced first in CTMs (Geer et al., 2006; Lahoz et al., 2007), and it is now well integrated also in operational CCMMs (Flemming et al., 2017). Today, numerous satellite retrievals of trace gases (e.g. $O_3$, CO, $NO_2$, $CH_4$, $CO_2$) and aerosols (AOD) are assimilated daily within operational CTMs and CCMMs (Inness et al., 2015; Bocquet et al., 2015). However, some aspects of the Data Assimilation (DA) approach differ between the chemistry and meteorology communities.

Since long time, meteorological variables such as temperature and water vapor profiles are corrected by means of assimilating directly satellites radiances (Level 1 data). Therefore, the RTM became part of the observation operator of the assimilation system (Andersson et al., 1994). This resulted necessary to avoid the introduction of biases in NWP that arise from poor prior information used in satellite retrievals at that time (Eyre et al., 1993). On the other hand, chemical species and aerosols are mostly corrected by means of assimilating geophysical retrievals (Level 2 or L2 data) that are made available by satellite data providers. To remove the impact of the prior information when assimilating L2 retrievals, the Averaging Kernels (AK) of the retrieval must be multiplied by the modeled profiles before computing the innovation vectors (Eskes and Boersma, 2003). However, within standard methods based on the linearization of the RTM, like the optimal estimation, issues might still arise when the prior information used in the retrieval sits far from the true atmospheric state: this might challenge the linearization of the observation operator and result in sub-optimal retrievals. Since the AK themselves are also a result of the retrieval (and depend upon its prior information), we suppose that a perfect removal of the prior information within DA cannot always be ensured.

The precise conditions that provide an equivalence between assimilating retrievals (using some kind of weighting functions) and radiances have been formalized by Migliorini (2012) and further tested by Prates et al. (2016) on synthetic satellite observations. These authors conclude that the equivalence holds under the hypothesis of an almost linear RT regime and with





a careful selection of the prior error covariances in a way to maximize the measurements information in the retrieval step. Nonetheless, testing the two approaches within an operational system and with real observations remains crucial to verify if these conditions are met in practice. Moreover, the perfect equivalence holds only when all the auxiliary inputs of the RTM are exactly the same in both the retrieval and the radiance assimilation. It is clear that a climatological option for some RTM

inputs will always be a more practical choice when computing L2 retrievals. On the other hand, the evolution towards strongly integrated ESMs will allow in principle to dispose of the most accurate prior information for all RTM inputs and favors the radiances assimilation approach. In this context, it appears important to introduce and evaluate the assimilation of radiances for chemical applications as well.

To the knowledge of the authors, the existent literature on this topic only concerned meteorological applications. Han and

McNally (2010) explored the possibility of assimilating $O_3$ sensitive radiances within a NWP model but without comparing the two approaches. Similarly, Weaver et al. (2007) examined the assimilation of satellite radiances for aerosols but the focus was on the impact of using modeled aerosols microphysical properties as auxiliary input for the RTM and no comparison was provided. No other studies could be found concerning the assimilation of chemical compounds.

The objective of this study is to perform a first strict comparison between radiances and retrievals assimilation, with re-

spect to $O_3$ estimation in the Thermal Infrared (TIR). To this end, systematic differences between the retrieval and radiances assimilation have been minimized as much as possible, for example by means of employing the same RTM within the two approaches.

We consider the case of $O_3$ assimilation using the Infrared Atmospheric Sounding Interferometer (IASI) onboard the European MetOP satellites (Clerbaux et al., 2009). Several IASI $O_3$ retrievals have been already well validated (Dufour et al., 2012)

and used directly to provide multi-annual time-series of the global $O_3$ budget (Wespes et al., 2016) or successfully assimilated within global (Peiro et al., 2018) and regional CTMs (Coman et al., 2012). However, an empirical correction of the retrievals has been found necessary to ensure globally unbiased reanalyses and slightly degraded assimilation results are still found at mid and high latitudes (Emili et al., 2014). Since the tropospheric $O_3$ signature in the selected IASI spectral window decreases over colder surfaces, the impact of the retrieval's prior might become more relevant at high latitudes. In addition, the majority

of IASI $O_3$ retrievals use a single a-priori profile globally (Barret et al., 2011; Boynard et al., 2016), which might present very large local departures from the true $O_3$ profile. Hence, IASI $O_3$ assimilation represents a good benchmark to evaluate the differences between retrievals and radiances assimilation.

The IASI-SOFRID $O_3$ product (Barret et al., 2011) and MOCAGE CTM have been used here to benefit from the experience of previous studies (Emili et al., 2014; Peiro et al., 2018). Both are based on a variational algorithm and, since SOFRID

employs RTTOV (Saunders et al., 1999), which is a community RTM developed originally for NWP applications, the same RTM has been implemented in the MOCAGE DA system. Global $O_3$ reanalyses are computed for July 2010 and the results are compared against all available radio-soundings to evaluate their accuracy. Since the sensitivity of IASI TIR measurements to $O_3$ is not uniform along the atmospheric column, we also investigate the impact of assimilating more accurate stratospheric profiles from the Microwave Limb Sounder (MLS) in combination with IASI radiances. This might reveal possible synergies

when assimilating multiple instruments that sense different layers of the atmospheres.





The paper is organized as follows. The satellite measurements, the Level 2 retrievals and the validation measurements used for this study are described in Sec. 2, as well as main steps concerning the preprocessing for some of the datasets. The chemical transport model, the radiative transfer model, the assimilation algorithm and the setup of the experiments are described in Sec. 3. The assimilation of IASI retrievals and radiances is compared in Sec. 4.1 and the impact of MLS assimilation on top of IASI

is discussed in Sec. 4.2. The conclusions are summarized in the last section, where some recommendations are also given.

## 2  Measurements

### 2.1  IASI

The Infrared Atmospheric Sounding Interferometer (IASI) flies onboard the series of polar-orbiting satellites MetOP operated by the EUropean organization for the exploitation of METeorological SATellites (EUMETSAT). It provides hyper-spectral

measurements of the Earth's thermal radiation in the 3.62 - 15.5 $\mu$m (2760 - 645 cm$^{-1}$) window and serves meteorological and atmospheric chemistry applications (Clerbaux et al., 2009). IASI is an operational mission meant to provide long-term (> 20 years) time series of accurate TIR spectra at high spatial resolution. A total of three IASI instruments will be flying simultaneously at the end of 2019, providing nearly global coverage three times per day. Hence, they represent a great opportunity for both NWP and climate-chemistry reanalyses. Only MetOP-A data, available from 2008 to present, have been employed for

this study.

#### 2.1.1  L1 radiances

IASI L1c data contain calibrated and geolocalized spectra at 0.5 cm$^{-1}$ spectral resolution (after apodization), i.e. 8700 radiance values for each ground-pixel, with a footprint of 12 km for nadir observations. For this study, historical L1c data granules have been downloaded from the EUMETSAT Earth Observation data portal (https://eoportal.eumetsat.int) in NETCDF format. Data

files contain also the observation geometry (sun and satellite angles) for each ground-pixel and the collocated land mask and cloud fraction values, obtained from the Advanced Very High Resolution Radiometer (AVHRR) measurements, also onboard MetOP.

#### 2.1.2  SOFRID L2 retrievals

The Software for a Fast Retrieval of IASI Data (SOFRID) was developed at the Laboratoire d'Aérologie to retrieve O$_3$ (Barret

et al., 2011) and CO (De Wachter et al., 2012) profiles from IASI in near-real time. It is based on the Radiative Transfer for TOVS (RTTOV) RTM (Saunders et al., 1999) and the 1D-VAR scheme developed within the Numerical Weather Prediction Satellite Application Facilities (NWP-SAF) program. SOFRID retrieves the O$_3$ profile in volume mixing ratio (vmr) units at 43 pressure levels between the surface and 0.1 hPa using 469 spectral channels within the main IASI O$_3$ window (980 - 1100 cm$^{-1}$). The choices that are made in SOFRID and are relevant for this study are summarized in Tab. 1. Note that a single

a-priori profile and error covariance matrix are used globally and that the Surface Skin Temperature (SST) is estimated within



the retrieval. The number of DOF of the SOFRID retrieval has been evaluated between 2 and 3 for the full atmospheric column, with about one DOF for the tropospheric column (Dufour et al., 2012). The accuracy of the retrieved $O_3$ depends on the latitude and the vertical level, but sits generally within 10-20 % of the corresponding radiosoundings values, once the averaging kernels are applied. However, increased biases are found in the troposphere with SOFRID (+10%) and positive biases of about 15%

are found in the Upper Troposphere - Lower Stratosphere (UTLS) region with all current IASI $O_3$ products (Dufour et al., 2012). The reasons for such biases are not yet fully understood and can impact negatively data assimilation (Emili et al., 2014) or trends analysis (Gaudel et al., 2018). This study will provide further insights about the impact of the constant a-priori on IASI retrievals. The SOFRID V1.5 retrievals described in Barret et al. (2011) are available for the full MetOP-A period at http://thredds.sedoo.fr/iasi-sofrid-o3-co . The V1.6 version of SOFRID retrievals has been used for this study and have been

obtained from LA (personal communication). The only difference with version 1.5 concerns the temperature and water vapor profiles employed in the radiative transfer computations, which are taken from the ECMWF NWP forecasts (V1.6) instead of EUMETSAT L2 retrievals (V1.5). Since the CTM is also based on ECMWF NWP forcing fields (Sec. 3.1), this choice minimizes possible systematic differences between L2 retrievals and L1 assimilation. In addition to the $O_3$ retrieval and its error covariance, SOFRID files contain a number of auxiliary and diagnostic fields. We considered in particular the SOFRID

cloud fraction, based on Brightness Temperature (BT) analysis at 11 and 12 $\mu$m to fill pixels with missing AVHRR data (Barret et al., 2011), and an index based on V-shaped sand signature computed as $\Delta BT = (BT_{829cm^{-1}} - BT_{972.5cm^{-1}}) + (BT_{1202.5cm^{-1}} - BT_{1096cm^{-1}})$. Usage of these products will be detailed in the data preprocessing section (2.4).

## 2.2 MLS L2 retrievals

Since 2004 The Microwave Limb Sounder (MLS) flies on-board the research mission AURA and measures thermal emission

at the atmospheric limb (Waters et al., 2006). It provides about 3500 stratospheric profiles of multiple atmospheric constituents each day, including $O_3$ (Froidevaux et al., 2008). Since the version 3 of MLS products, $O_3$ profiles are retrieved on 55 pressure levels with a recommended range for scientific usage between 0.02 and 261 hPa for version 4.2 (Livesey, 2018). The biases of MLS $O_3$ profiles are typically within 5% with respect to ozonesondes and lidar measurements (Hubert et al., 2016), with slightly higher values below 200 hPa. Given its good accuracy, MLS $O_3$ has been widely used both for trend analysis (Froidevaux et al.,

2015) and assimilation experiments (Massart et al., 2010; Miyazaki et al., 2012; Inness et al., 2015). Similarly to previous studies (Emili et al., 2014), we retain only the most accurate data using MLS, i.e. above 170 hPa. The MLS V4.2 product used in this study has been downloaded from the Goddard Earth Sciences Data and Information Services Center (GES DISC) web portal (https://disc.gsfc.nasa.gov).

## 2.3 Radiosoundings

Ozonesondes are launched on weekly bases by meteorological services and provide accurate profiles of $O_3$ up to 10 hPa with a vertical resolution of 150-200 m. ECC type sondes, which represent the largest percentage of the global network, have a precision of about 5% (Thompson et al., 2003). Radiosoundings are relatively sparse and their geographical distribution is much more representative of the northern mid-latitudes. However, they provide since several decades the most precise



information on vertical ozone distribution in the troposphere. Therefore, they have been used to derive widely used tropospheric $O_3$ climatologies (McPeters et al., 2007) and validate both satellite products (Dufour et al., 2012) or models (Geer et al., 2006). They will be used in this study to validate all model simulations. Data are collected and distributed by the World Ozone and Ultraviolet Radiation Data Center (WOUDC, http://www.woudc.org).

## 2.4 Data preprocessing

Some further preprocessing has been applied to the original L1c and SOFRID datasets to ease the interpretation of the assimilation experiments presented later in Sec. 4. The objective was to ensure that exactly the same spectra are used for both L1 and L2 assimilation.

Only the spectral channels that are used in SOFRID are extracted from IASI L1c granules, i.e. channel n. 1350 (980 cm$^{-1}$) to 1818 (1100 cm$^{-1}$). Some further screening is applied to remove channels that are affected by strong $H_2O$ absorption, as also done in SOFRID.

The spatial resolution of the CTM (2°x2° degrees, Sec. 3.1) is much coarser than IASI pixels size. Since it is preferable to avoid all kind of spatial averaging of the observations, a significant reduction of ground-pixels is needed. In return, we employ strict selection criteria to avoid as much as possible contamination from clouds and bright surfaces, which reduce the RT accuracy and increase retrieval or assimilation errors. The data selection is performed as follows.

First, only L1 pixels with both IASI and AVHRR highest quality flags are kept. Then, ground-pixels from IASI L1 and SOFRID products are filtered using their respective cloud masks (Sec. 2.1.1 and 2.1.2) and keeping only pixels with cloud fraction less or equal to 1%. SOFRID pixels with a sand signature greater than 0.5 and with a number of retrieved levels lower than 35 (mountains) are also filtered out. Resulting datasets are then matched, i.e. only common ground-pixels that remained available after the previous L1 and SOFRID independent selections are kept. Finally, data thinning is performed to retain a maximum of about two pixels for each model grid point. After the completion of the data selection procedure the final number of retained ground-pixels for L1 and SOFRID is about 3300 per day, compared to about $10^5$ when only the cloud screening is applied.

## 3 Method

This section summarizes the main characteristics of the CTM (3.1), the RTM (3.2) and the assimilation algorithm (3.3) used in this study. Further details on the particular selection of the main parameters of the assimilation experiments (e.g. the error covariances) are given in Sec. 3.4.

## 3.1 Chemical transport model

The Chemical Transport Model (CTM) MOCAGE (Josse et al., 2004) is used in this study. A global configuration with an horizontal resolution of 2°x2° degrees and 60 hybrid sigma-pressure levels up to 0.1 hPa has been used. The vertical resolution varies from about 100 m in the planetary boundary layer to about 700 m in the upper troposphere, decreasing further to





approximately 2 km in the upper stratosphere. Chemical mechanism, emissions and physical parameterizations follow the setup used for operational air-quality forecasts (Marécal et al., 2015), which includes about 100 species and 300 chemical reactions. A similar configuration has been employed by Barré et al. (2013) to assimilate IASI $O_3$ columns over Europe, but with a lower model top at 5 hPa. Other authors favored a simplified chemistry scheme but with a model top at 0.1 hPa to

assimilate satellite $O_3$ products globally (Emili et al., 2014; Peiro et al., 2018).

We considered for this study the highest available model top because we need to simulate the full atmosphere to compute radiances. In addition, the 0.1 hPa top matches with the vertical grid used for SOFRID retrievals (Sec. 2.1.2), making the comparison of the two assimilation approaches (radiances vs L2) stricter. The full chemical scheme is chosen instead of a simplified chemistry to reduce as much as possible biases of the modeled $O_3$ in the troposphere. The main intent of this study

is in fact to evaluate the impact of a dynamical and accurate $O_3$ prior on assimilation results.

The meteorological forcing comes from the ECMWF NWP model (IFS), from which we selected the forecast steps initialized with the latest available analysis (at 00 or 12 UTC).

## 3.2 Radiative transfer model

RTTOV (Saunders et al., 1999) is a community RTM developed for operational NWP models. One of its main advantages is

computational efficiency, which is achieved by running accurate but costly line-by-line RT simulations for a large number of satellite sensors, observation geometries and atmospheres and storing the corresponding coefficients in large look-up tables. RTTOV provides API interfaces for the direct RT computations plus the tangent linear and adjoint model, which are needed in variational assimilation systems.

Version 11.3 of RTTOV (Saunders et al., 2013) has been used in this study. This version includes coefficients for the IASI

TIR channels computed using a fine atmospheric grid (104 vertical levels). The SST, 2 m temperature, 2 m pressure and 2 m wind vector are taken from high resolution (0.125°x0.125° degrees) global IFS forecasts initialized from ECMWF 4D-Var analysis at 00 UTC of each day and collocated with satellite ground-pixels prior to data assimilation. The surface emissivity is based on the RTTOV monthly TIR emissivity atlas (Borbas and Ruston, 2010). Only clear-sky RT computations are performed for this study and no aerosols have been prescribed. The RTM configuration is summarized in Table 1.

## 3.3 Assimilation algorithm

The assimilation suite for MOCAGE is based on a variational algorithm and was developed initially within the ASSET (Assimilation of Envisat data) project (Lahoz et al., 2007). The objective was to assimilate satellite products at a global scale and a 3D-FGAT implementation was chosen. It evolved later to provide air-quality reanalyses at the surface based on a 3D-Var implementation (Jaumouillé et al., 2012) and extended to 4D-Var in case of linearized chemistry schemes (Massart et al., 2012;

Emili et al., 2014). In all cases the minimization of the variational cost function is performed using the limited-memory BFGS algorithm (Liu and Nocedal, 1989). We used in this study a 3D-Var algorithm with hourly assimilation windows and with $O_3$ as control variable.




The 3D background error covariance is modeled through a diffusion operator (Weaver and Courtier, 2001) and allows the specification of heterogeneous correlation length scales. Compared to previous studies using MOCAGE assimilation suite, a new vertical correlation operator has been employed here: the vertical error correlation is now assigned by explicitly filling a positive definite matrix using the gaussian formulation of Paciorek and Schervish (2006) and by numerically computing its

square root. This avoids difficulties encountered with diffusion based operators concerning the normalization in presence of boundaries (e.g. the surface) and heterogeneity (Mirouze and Weaver, 2010). Since the vertical dimension of the model grid is relatively small, this choice does not impact significantly the numerical cost and the memory requirements with respect to the previous implementation based on diffusion.

The observation operator of MOCAGE allows to assimilate a large number of measurements, spanning from columns of

gases (Massart et al., 2009) to aerosol optical depth (Sič et al., 2016). Next, we give some details of the implementation used in this study to assimilate vertical profiles and radiances.

After the horizontal and temporal interpolation of the model fields at the satellite ground-pixel position, modeled profiles are linearly interpolated to the retrieval's vertical grid. When the averaging kernels are used (i.e. for SOFRID assimilation), the linear estimation equation (Barret et al., 2011) is used to remove the impact of the prior from the innovation vector. The

ensemble of these operations is stored as coefficients of a large sparse matrix and done through its multiplication by the model 3D field. This approach is practical since numerous application of the linearized and adjoint operator are needed during the minimization of the variational cost function. Differently from all previous studies involving IASI $O_3$ assimilation (Massart et al., 2009; Emili et al., 2014; Peiro et al., 2018), where L2 profiles were first reduced to total or partial columns prior to assimilation, we assimilate here directly the full L2 profiles (43 levels). This avoids any loss of information and allows a fairer

comparison between L2 and radiances assimilation. The error covariance matrix of the profile-type observations is diagonal in the latitude/longitude dimensions but off-diagonal terms are allowed along the vertical dimension.

The steps for the computation of modeled radiances are equal to the profiles ones until the vertical interpolation. In fact, the RTTOV vertical interpolator is used for radiances computations instead of the MOCAGE one. All model levels (60) and corresponding levels pressure are given as input to RTTOV, which performs internally the vertical interpolation to the IASI

coefficients levels. Since the model vertical resolution is lower than the one available in RTTOV for IASI coefficients (104 levels), we used the default option based on Rochon et al. (2007). Also, $O_3$ profiles above the CTM top (0.1 hPa) are completed using RTTOV climatological profile. Auxiliary inputs for the radiances computation include the pressure, temperature and water vapor profiles, which are interpolated from the correspondent MOCAGE fields.

The MOCAGE control vector has been extended to include the SST, as it is done within SOFRID retrieval scheme. This

proved to be important since small errors in the SST translate in significant differences between modelled and measured radiances. Not accounting for this would produce wrong $O_3$ analyses. The SST does not belong to the MOCAGE prognostic fields nor it is prescribed on the MOCAGE grid. Hence, the SST analysis is not propagated in time and no spatial covariance model have been implemented so far. In this sense, it can be interpreted as a variational bias correction term in the observation space (Dee and Uppala, 2009), with prior values given by the NWP model (IFS, see Sec. 3.2).



### 3.4 Setup of the experiments

We performed numerical experiments for the month of July 2010, which corresponds to the typical presence of summer $O_3$ maxima in the northern hemisphere linked to photochemical pollution. July 2010 is also interesting due to the development of a strong La Nina episode (Peiro et al., 2018). The main difference between assimilating L2 and L1 data consists in using

a climatological (L2 assimilation) versus a dynamical a-priori (L1 assimilation) for the inversion of the radiative transfer problem. The chosen period presents large local deviations of the $O_3$ field from climatological values. Therefore, it provides an interesting benchmark period with respect to the objective of this study.

The CTM has been initialized on 1st June 2010 with a zonal climatology and run for one month period (spinup) to provide chemically balanced initial condition on 1st July 2010 for all simulations.

The observation error covariance matrix ($\mathbf{R}$) is prescribed according to the choices adopted in SOFRID V1.6. When the radiances are assimilated, a diagonal matrix (i.e. with no inter-channel correlation) is used with a constant standard deviation of to 0.7 mW m$^{-2}$ sr$^{-1}$ cm for all channels. This is a simplified although common setting for most IASI $O_3$ retrievals (Barret et al., 2011; Boynard et al., 2016). The SST, which is controlled as well within radiances assimilation, has a prescribed standard deviation of 4° C for all ground-pixels. When L2 profiles are assimilated we used the full non-diagonal error covariance matrix

provided by SOFRID or MLS retrievals.

We considered a dynamical rejection of observations based on the relative differences between simulated and measured values with respect to simulated values. It avoids assimilating observations with too large departures from corresponding model background. The thresholds values are set to 12% for L1 radiances and 2000% for L2 profiles and trespassing the threshold for any particular channel or profile level rejects the entire spectrum or profile. The strong difference between the two thresholds

is a consequence of the very different nature of assimilated observations: the exponential shape of $O_3$ profiles can produce very large departures where the gradient is the steepest (tropopause) and a small rejection threshold would filter out most of the profile observations. This is not the case for radiances, which vary on a linear scale. Thresholds values have been chosen based on misfit histograms in a way to remove abnormal tails. As a consequence, L1 and L2 pixels that pass the selection and are further assimilated could differ. However, the relative number of rejected observations for the entire month of July is quite

limited in both cases (3% for L1, 6% for L2), thus not affecting statistically the results.

The setup of the background error covariance ($\mathbf{B}$) is a critical step both for L2 retrievals and data assimilation. For this study we could benefit from past experiences using MOCAGE, IASI and MLS $O_3$ (Massart et al., 2012; Emili et al., 2014; Peiro et al., 2018) and we tried to derive an optimal parameterization for $\mathbf{B}$ based on previous results. Note that the $\mathbf{B}$ matrix (3D) used in data assimilation is by definition different with respect to the one specified within SOFRID (1D), but the same 3D $\mathbf{B}$

is used for all data assimilation experiments (L1 and L2). Concerning the standard deviation, Emili et al. (2014) employed vertically varying errors expressed as percentage of the background $O_3$ profile. Optimal results were found setting a value of 5% in the stratosphere and 30% in the troposphere, with the tropopause being arbitrary set at about 150 hPa. Peiro et al. (2018) kept the same error parameterization but reduced the errors to 15% in the troposphere to analyze the tropical $O_3$ distribution. Since we use here a more detailed chemistry model (Sec. 3.1) we first evaluated the Root Mean Square Error (RMSE) of a



free model simulation (control) against the ozonesondes (Fig. 1). We remark that the model's RMSE reproduces the vertical features observed in previous studies, with smaller errors in the stratosphere (between 20 and 50 hPa), larger errors in the free troposphere, and highest errors close to the tropopause and within the planetary boundary layer. Note also the zonal variability of the maxima, which appear linked to the variability of the tropopause height. However, thanks to the detailed chemical

mechanism, biases (not shown) are generally smaller than in the studies cited previously.

The background standard deviation is prescribed through a smooth step function that takes values of 2% above 50 hPa and 10% below. These values are slightly smaller than in previous studies (Emili et al., 2014; Peiro et al., 2018) because of the smaller biases of the forecast model. Also, the percent profile is multiplied by the hourly $O_3$ field of the control simulation once for the entire period and not at every forecast time step. Therefore, all assimilation experiments presented in this study

are based on the same $\mathbf{B}$ matrix. This choice has been taken to permit a stricter comparison between L1 and L2 assimilation experiments.

The vertical error correlation diffuses the assimilation increments between model levels and has been found to significantly impact the quality of $O_3$ reanalyses with current model vertical resolutions (not shown). In general, small values of vertical correlation are favored to avoid injection of large stratospheric $O_3$ increments in the troposphere. For example, Emili et al.

(2014) used a constant correlation length of 1 model grid point; Peiro et al. (2018) found that switching off the vertical correlation provided even better results for MLS analyses. However, a non-zero correlation seems more appropriate for generic usage, because it allows to assimilate effectively also point measurements. Second, the SOFRID prior covariance is also non-diagonal (Barret et al., 2011) and it is better to preserve a certain consistency between the two approaches. Therefore, we used the value of 1 model grid point in this study.

Finally, the exponential scale of the horizontal error correlation is set equal to 200 km, with the zonal component that is reduced towards the poles to account for the increasing resolution of the model's grid (Emili et al., 2014).

Further improvements of the $\mathbf{B}$ parameterization could be achieved by diagnosing the forecast errors hourly or using ensembles of model forecasts. However, such complex and costly estimations do not always improve systematically and significantly the results of chemical assimilation (Massart et al., 2012). Additional research is needed in this regard, which is out of the

scope of this study.

## 4   Results

A total of six simulations for the month of July 2010 have been performed (Tab. 2), starting on 1st July: a free model simulation (control) and five 3D-Var reanalyses assimilating respectively SOFRID L2 profiles (named L2a), IASI L1 radiances (L1a), MLS L2 profiles (MLSa), MLS plus SOFRID L2 profiles (MLS+L2a), MLS plus L1 radiances (MLS+L1a). The first three

simulations (control, L2a and L1a) are discussed in Sec. 4.1. The control simulation and the three reanalyses that include MLS are discussed in Sec. 4.2. All simulations have been validated against ozonesondes profiles to elucidate the differences of the resulting $O_3$ vertical distribution. A total of 220 radiosoundings are available globally in July 2010. The colocation of ozonesondes profile with model fields in time and space is performed through the MOCAGE observation operator (Sec. 3).



### 4.1 IASI assimilation

We discuss the geographical differences between L1a and L2a reanalyses by looking at the monthly bias between the two experiments, divided by the average $O_3$ of the control simulation. To this end $O_3$ fields have been first interpolated vertically from the model grid to a selection of pressure levels, covering both the stratosphere and the troposphere, and averaged afterwards.

Relative differences are displayed in Fig. 2. First, we remark that differences are in generally significant both in the stratosphere and in the troposphere, with absolute values that can exceed 50% of the $O_3$ field locally and global averages that are between 1% and 15%. Largest differences in the stratosphere are found in correspondence of tropical latitudes, L1a showing larger $O_3$ values than L2a at 20 hPa and lower at 70 hPa. In the troposphere the strongest positive differences are still found in the tropics, especially over central Africa and Eastern Asia, but significant negative differences appear in the Southern Hemisphere (SH)

mid latitudes. Differences become smaller when moving down to 750 hPa and tend to disappear at lower altitudes (not shown), which is normal considering the vertical sensitivity of IASI. More remarkably, in the Northern Hemisphere (NH) mid and high latitudes, relative differences are smaller than elsewhere. This behavior seems coherent with the fact that the SOFRID prior is more representative of the NH mid-latitudes (Sec. 2.1.2) and much less accurate for tropical and SH latitudes. Overall, these plots suggest that the equivalence between L1 and L2 assimilation is not verified for $O_3$, even when the averaging kernels are

employed.

To further verify which one, between the L1a and L2a experiments, reproduces better the measured $O_3$ profiles, we validated the three simulations against radiosoundings. Figure 3 reports the RMSE differences computed globally and for five different latitude bands. The displayed values are the differences between the RMSE of the assimilation experiment and the corresponding value for the control simulation (Fig. 1). Negative values in Fig. 3 indicate that the assimilation improved the

$O_3$ field and decreased the relative RMSE with respect to ozonesondes by the amount displayed on the plot. Looking at the global averages we remark that below 70 hPa the gain is similar for both L1a and L2a experiments, and quite significant at 200 hPa (20%). Note, however, the strong similarity between the global and 30°N-60°N statistics, due to the over-representation of ozonesondes for NH mid-latitudes (63% of the total).

In the NH the RMSE of the control simulation is effectively reduced between 70 and 300 hPa (up to 20%). L1a shows a

slightly better gain than L2a between 150 and 300 hPa. Interestingly, both L1a and L2a display increased RMSE between 300 and 400 hPa. This behavior is also confirmed when the vertical error correlation is switched off in the 3D-Var **B** and with different choices for the vertical interpolation of $O_3$ optical coefficients within RTTOV (log-linear or Rochon, not shown). Since large negative biases were present in the control simulation (as low as -30%, not shown), a possible explanation is that part of the strong positive correction of $O_3$ between 100 and 300 hPa is propagated downwards, where both absolute

$O_3$ concentrations and relative biases are much lower. This can degrade the reanalysis accuracy below 300 hPa. Whether this propagation is carried out by the Jacobian matrix of the observation operator (either through the RTM or the retrieval's AK) or by vertical $O_3$ transport is not yet elucidated and would need further investigation. Also, other possible factors affecting the accuracy of the RTM exist, like inadequate vertical resolution close to the tropopause, uncertainties in meteorological profiles or impact of aerosols. Nonetheless, these errors impact both L1a and L2a in our study: a more profound revision of the L1





assimilation configuration with respect to the L2 retrievals is left for a future study. The RMSE is reduced again at about 500 hPa between 30°N-60°N, although not very significantly. The assimilation increases the RMSE of the tropospheric profile at northern latitudes (60°N-90°N). In general, the validation confirms that L1a and L2a have a very similar accuracy in NH at mid and high latitudes, as also suggested previously by Fig. 2. However, the strongest positive corrections are confined to the

UTLS.

At the tropics (30°S-30°N) the results differ more significantly. In the troposphere (below 100 hPa), both L1a and L2a reduced the RMSE of the control simulation, although by a smaller amount than in NH (5%). Note also that L1a RMSE reduction is larger than L2a between 300 and 500 hPa, whereas it is the other way round at about 600 hPa. Above 100 hPa we observe an increase of RMSE that peaks at 60 hPa with L2a and at 30 hPa with L1a, but smaller in magnitude for L1a. This

behavior might be linked to the strong differences that exist between the SOFRID prior and the modeled $O_3$ at the tropical tropopause, to some other factor affecting the RT computations, to overestimation of the background error covariances or to a complex combination of all previous causes. A full satisfactory explanation has not been found yet.

Results in the SH (30°S-90°S) are in favor of L1a: lower RMSE than for the control simulation is found for both L1a and L2a in the stratosphere (between 30 hPa and 100 hPa), with L1a slightly better at polar latitudes (60°S-90°S). More noticeably,

L2a is equal or worse than the control simulation in the troposphere (below 250 hPa), whereas L1a improves the RMSE. The SOFRID prior is biased towards NH mid-latitudes, where tropospheric $O_3$ concentrations are generally the highest. The sensitivity of IASI TIR channels to tropospheric $O_3$ decreases over colder surfaces (e.g. in the SH during July). Hence, we expect a stronger impact of the prior in the retrieval results, which can be detrimental if the prior is biased. Indeed, we note that L1a remains close to the (already accurate) control profiles, whereas L2a adds a positive bias (not shown). Such behavior

was already diagnosed by Emili et al. (2014) when assimilating SOFRID partial columns and we provide here a possible explanation. Using a more adapted prior in the SH could in principle also improve L2 retrievals themselves, which seems the case with a newer versions of SOFRID (B. Barret, personal communication).

Since radiosoundings do not provide a uniform global coverage and vertical coverage also lacks in the vicinity of the $O_3$ maximum, we validated the three simulations against MLS measurements. The RMSE differences for stratospheric profiles can

be found in Fig. 4. These statistics are based on more than $10^5$ profiles for the global average and between 15000 and 30000 for zonal averages, depending on the latitude band. The patterns observed in the stratosphere with respect to ozonesondes are confirmed also with MLS. The only exceptions are a lower RMSE degradation at 50 hPa for L2a in the tropics and for both L1a and L2a at 150 hPa in the 30°S-60°S band. Overall, the validation against MLS bolsters the robustness of the conclusions derived previously for the troposphere.

The computational cost of L1 assimilation is necessarily higher than for L2 assimilation. Additional CPU time is due not only to online RTM computations but also to a higher number of iterations needed by the minimizer to converge. For a typical 24 hours long simulation performed on Intel Xeon E5-2680 V3 CPU the total computing time is 3.9 CPU hours for L2a and 13.2 hours for L1a. Note that the L2a time does not include the cost of the L1 to L2 processor but only the cost of the 3D-Var assimilation plus the model forecasts. Most of the CPU time for L1a is spent in the linearized and adjoint calls of the RTM

(50% of the total CPU time), whereas the corresponding time spent for the observation operator within the L2a experiment is



about 1%. However, the total CPU time can be significantly decreased by reducing the maximum number of iterations of the minimizer. A simulation with halved number of iterations (75) showed very similar results to the ones that have been reported (150 iterations) and could be considered if computation time is a critical factor. Moreover, with standard high performance computers and thanks to the parallel nature of the observation operator and the RTM, we could obtain a speedup of about 24

on the 24 CPU cores. This reduces the run time of L1a to about 36 minutes for the 24 hours-long simulation, versus 13 minutes for L2a. The extra cost of L1 assimilation seems therefore acceptable also for operational applications.

### 4.2  IASI and MLS assimilation

Some issues were identified in the previous section in the stratosphere, especially at tropical latitudes. Among possible reasons, one is that inversion of TIR measurements might be particular sensitive to the vertical distribution of $O_3$ in the tropical

stratosphere. We consider here assimilating MLS L2 profiles on top of IASI to correct the model stratosphere and troposphere simultaneously, as done also in previous studies (Emili et al., 2014; Peiro et al., 2018). When the radiances are assimilated, the RT problem is solved for the entire atmospheric column within the iterations of the variational algorithm. Therefore, enhanced and better synergies could be observed than when only L2 products are assimilated.

We report in Fig. 5 the impact of assimilating MLS alone by computing the average difference between MLSa and the

control (upper plots), and the impact of assimilating MLS on top of IASI L1/L2 by computing the average differences between MLS+L1a (MLS+L2a) and L1a (L2a) respectively. As expected, the impact of MLS assimilation is very significant at 100 hPa (relative differences as high as 60% of control $O_3$) but becomes minor at 500 hPa, where no direct constraint exists from the observations. Interestingly there are regions at mid-latitudes where the impact of MLS is not negligible (> 5%). Since the 3D-Var increments are confined to higher levels (above 200 hPa), we reckon that the impact of MLS assimilation at 500 hPa is

due to the model dynamics at mid-latitudes, e.g. Stratosphere-Troposphere Exchanges (STE).

When comparing the MLS impact at 500 hPa with the bottom plots we remark that there is no sign of a strong spatial correlation in the NH and in the tropics. This suggests that the impact of MLS in the troposphere is supplanted by IASI assimilation (either L1 or L2), which is expected due to the strong sensitivity of IASI TIR measurements at 500 hPa. Traces of superposition of the MLS impact on IASI reanalyses appear in SH mid-latitudes, which is coherent with the fact that the

IASI tropospheric impact is smaller over colder surfaces (Sec. 4.1). In case of no synergy between MLS and IASI we would expect to see in bottom plots either very small values or patterns similar to what observed in the SH mid-latitudes. Instead, significant differences (as high as 10%) arise at tropical latitudes, which are also opposed in sign, i.e. a positive feedback of MLS is observed within MLS+L2a, both negative and positive, but smaller in amplitude, within MLS+L1a. This confirms that constraining the model with MLS above 200 hPa has a significant impact on the free troposphere when assimilating IASI.

We compared the RMSE of MLSa, MLS+L1a and MLS+L2a against ozonesondes (Fig. 6) to evaluate if some of the observed feedbacks improve the $O_3$ distribution. MLSa provides particularly accurate results down to 200 or 300 hPa, depending on the latitude, with a robust reduction of the RMSE with respect to the control simulation. The only exception is in the SH mid-latitudes below 250 hPa, where the MLSa RMSE increases. We suspect that this might be linked again to the combination of strong $O_3$ gradients at the tropopause height and the negative bias of the control simulation above the tropopause (see Sec.





4.1). Overall MLSa confirm results found in past studies (Massart et al., 2012; Emili et al., 2014) and represents a much better prior for assimilation of radiances or retrievals.

We remark that MLS+L1a and MLS+L2a provide now closer results in the NH and in the tropics compared to Fig. 3. This suggests that the small differences found previously between L1a and L2a in the NH (Fig. 2) were mostly due to the impact of the stratospheric $O_3$ on the radiative transfer computations. The stratospheric $O_3$ gain is much more significant with MLS+L1a/MLS+L2a than with L1a/L2a and remains very close to MLSa, demonstrating that assimilating accurate stratospheric profiles remains essential for $O_3$ reanalyses. In the NH, a positive, albeit small, effect of assimilating IASI on top of MLS is found between 150 and 300 hPa. Below 300 hPa, the addition of MLS does not bring further improvements with respect to IASI alone. Significant differences persist in the tropical troposphere and in the 30°S-60°S band, where MLS+L1a shows improved RMSEs with respect to MLS+L2a. In particular, only the assimilation of radiances allows to partially mitigate the RMSE degradation due to MLS in the SH (30°S-60°S) troposphere.

We report in Fig. 7 the Taylor plots concerning the free troposphere $O_3$ column (340-750 hPa), to further evaluate the skills of the assimilation experiments in terms of variability. We examine here the free troposphere since it is where the direct impact of IASI assimilation is the largest and the impact of MLS the smallest (except for the 30°S-60°S band). IASI assimilation improves the variability of the modeled $O_3$ field when looking at global averages, but this conclusion varies as a function of the latitude band. Robust and significant improvements are found only at the tropics and in the SH polar region, mixed elsewhere. This confirms previous findings obtained with L2 assimilation (Emili et al., 2014) and adds the conclusion that a better prior does not necessarily solve all issues related to the assimilation of TIR measurements at high latitudes. Nevertheless, the assimilation of radiances provides in general slightly better results at all latitudes and permits to extract more variability from IASI spectra especially at tropical latitudes.

## 5 Conclusions

In this study we addressed the following question: which are differences between the direct assimilation of IASI radiances (Level 1) and the assimilation of Level 2 products for $O_3$ reanalyses. We used an experimental setup where differences between the L2 retrieval and the assimilation algorithm have been minimized as much as possible, for example by using the same RTM (RTTOV) and control vector ($O_3$ and SST) in both approaches. This allowed to delve into the impact of the $O_3$ prior and its error covariance on the quality of the reanalysis.

We performed twins assimilation experiments with the MOCAGE CTM and the SOFRID $O_3$ retrievals, using the same IASI ground-pixels for both L1 and L2 assimilation, named L1a and L2a respectively. We compared the obtained reanalyses between each other and against ozonesondes for the month of July 2010.

Main findings suggest that the accuracy of the $O_3$ prior information used in the L2 retrievals can influence significantly the assimilation results. When the $O_3$ prior is biased and the sensitivity of the retrieval is small (e.g. in the SH troposphere in winter) increased errors with respect to the control simulation are found assimilating L2 profiles (with the respective kernels). When the sensitivity is larger, but the retrieval's prior is still biased (in the tropical troposphere), the reanalysis shows a better





variability when assimilating directly L1 radiances instead of L2 profiles. L1a and L2a are instead very similar in the NH, where the SOFRID $O_3$ prior is the closest to the truth.

We conclude that particular care should be taken before assimilating satellite retrievals that in some circumstances can have a low sensitivity to the true profile. A thorough analysis of the retrieval's DOF and averaging kernels represents the first step
in this direction, but the dependence of these diagnostics to the prior itself can make this analysis troublesome. Assimilating directly L1 radiances represents a viable alternative to this. We could imagine extending this analysis to other chemical species (e.g. CO) and spectral regions (e.g. UV) that show a similar behavior to $O_3$ in the TIR spectrum in terms of information content of the measurements.

Finally, a positive synergy has been found when assimilating simultaneously MLS profiles and IASI (either L1 or L2),
which corrected stratospheric biases due to IASI assimilation alone. The addition of MLS was found to influence the results of IASI assimilation also in the free troposphere (500 hPa), with L1 assimilation providing in general better results than L2 in the tropics and in the SH. This suggests that using L1 data might also be beneficial in a context of assimilating multiple instruments with different vertical sensitivities at the same time.

We reckon that L1 assimilation requires modeling the full atmosphere, which may be not available to some models, those
for example conceived exclusively for tropospheric applications. Moreover, Level 2 products can be aggregated vertically to correct selectively some model layers and averaged spatially to fit models with coarser resolution than the satellite ground-pixel size. This cannot be easily done with radiances and should be addressed in future research.

In this study the observations, their error covariance and the RTM auxiliary inputs were kept almost identical between L1 and L2 assimilation on purpose. Further research is needed to address issues that are common to L1 and L2 assimilation, e.g.
increased errors close to the tropopause in the NH or in the tropical stratosphere. Improvements are expected for example by increasing the vertical resolution of the model, including modeled aerosols within the RT or using more realistic observation error covariances. Including more modeled variables among the RTM inputs is in particular of interest in the context of the evolution towards ESMs, where hyper-spectral sounders like IASI can provide very valuable constraint for multi-variate re-analyses (atmosphere plus surface). Including inter-channel and ground-pixel correlations in the observations error covariance
matrix seems necessary to correctly weight IASI very dense observations within higher resolution models than the one used in this study. All these aspects deserve further research.

*Competing interests.* The authors declare that they have no conflict of interest.

*Acknowledgements.* We acknowledge EUMETSAT for providing IASI L1C data, WOUDC for providing ozonesondes data and the NASA Jet Propulsion Laboratory for the availability of Aura MLS Level 2 $O_3$. We also thanks the MOCAGE team at Météo-France for providing
the chemical transport model, the RTTOV team for the radiative transfer model, Andrea Piacentini and Gabriel Jonville for the help on technical developments of the assimilation code. This work has been possible thanks to the financial support from the Région Midi-Pyrénées,



who sponsored the preliminary work of Hélène Peiro on the subject, and CNES (Centre National d'Études Spatiales), through the TOSCA program.





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




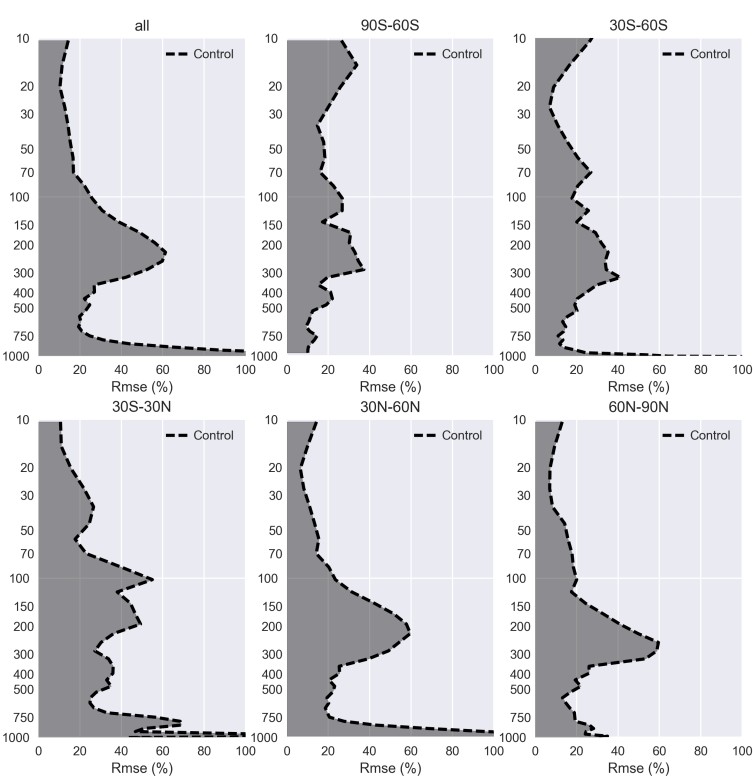

**Figure 1.** Relative Root Mean Square Error (RMSE) of the control simulation with respect to radiosoundings averaged globally (first plot) and for five latitude bands separately (90°S-60°S, 60°S-30°S, 30°S-30°N, 30°N-60°N, 60°N-90°N).





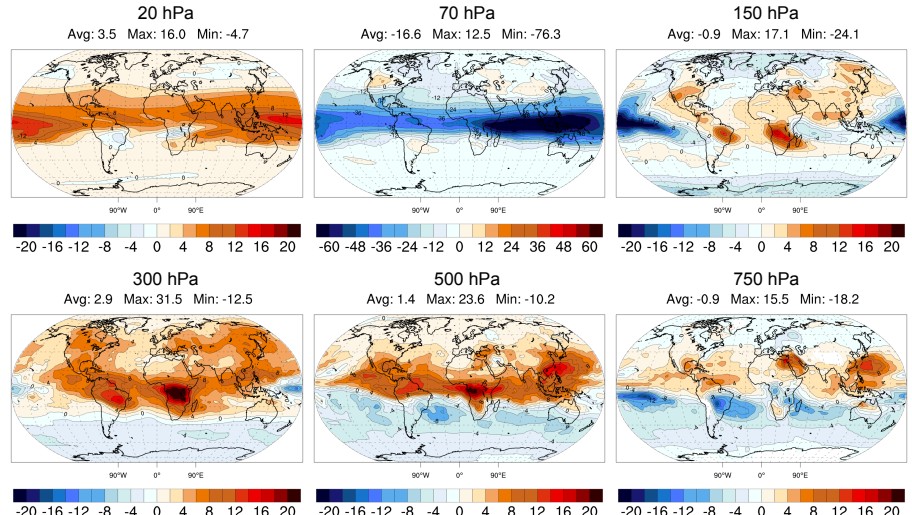

**Figure 2.** Relative differences (%) between radiances and Level 2 assimilation (L1a minus L2a divided by the correspondent $O_3$ values of the control simulation) averaged on July 2010. From left to right different pressure levels are displayed covering the stratosphere (top) and the free troposphere (bottom). Average, maximum and minimum values of the displayed fields are given on top of each map.

**Table 1.** Summary of the configuration of SOFRID L2 retrievals and MOCAGE L1 assimilation.

|  | L2 retrieval | L1 assimilation |
| --- | --- | --- |
| Radiative transfer model | RTTOV V9 | RTTOV V11.3 |
| Algorithm | 1D-Var | 3D-Var |
| Spectral window | 980 - 1100 cm$^{-1}$ | 980 - 1100 cm$^{-1}$ |
| Measurements error | 0.7 (mW m$^{-2}$ sr$^{-1}$ cm) | 0.7 (mW m$^{-2}$ sr$^{-1}$ cm) |
| Control vector | $O_3$ (1D) + Surface Skin Temperature (SST) | $O_3$ (3D) + Surface Skin Temperature (SST) |
| Vertical grid | 43 pressure levels (1013-0.1 hPa) | 60 hybrid sigma-pressure levels (surface-0.1 hPa) |
| $O_3$ prior | MLS+Ozonesondes global climatology | 3D-hourly model forecasts |
| $O_3$ error covariance | MLS+Ozonesondes climatological covariance | 3D-hourly (standard deviation), parameterized (correlations) |
| SST prior | ECMWF-IFS operational forecasts | ECMWF-IFS operational forecasts |
| SST error covariance | 4°C | 4°C |
| Temperature, water vapor | ECMWF-IFS (on 43 levels) | ECMWF-IFS (on MOCAGE 60 levels) |
| IR emissivity | (Borbas and Ruston, 2010) | (Borbas and Ruston, 2010) |




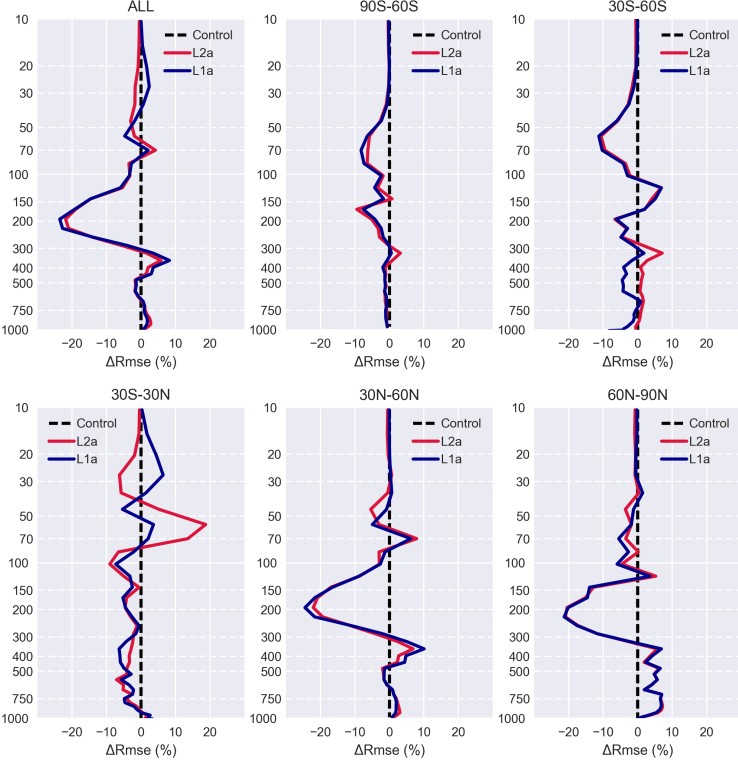

**Figure 3.** Relative difference of RMSE (ΔRMSE) with respect to radiosoundings for L1a (blue) and L2a (red). The difference is computed by subtracting the RMSE of L1a (L2a) against radiosoundings from the RMSE of the control simulation (Fig. 1). Negative values mean that the assimilation improved (decreased) the RMSE of the control simulation, positive values indicate degradation (increase) of the RMSE. The statistics are computed for the same latitudes as in Fig. 1.

**Table 2.** Names of experiments and assimilated data.

| Experiment's name | IASI L1 | IASI L2 | MLS L2 |
|---|---|---|---|
| Control | no | no | no |
| L1a | yes | no | no |
| L2a | no | yes | no |
| MLSa | no | no | yes |
| MLS+L1a | yes | no | yes |
| MLS+L2a | no | yes | yes |





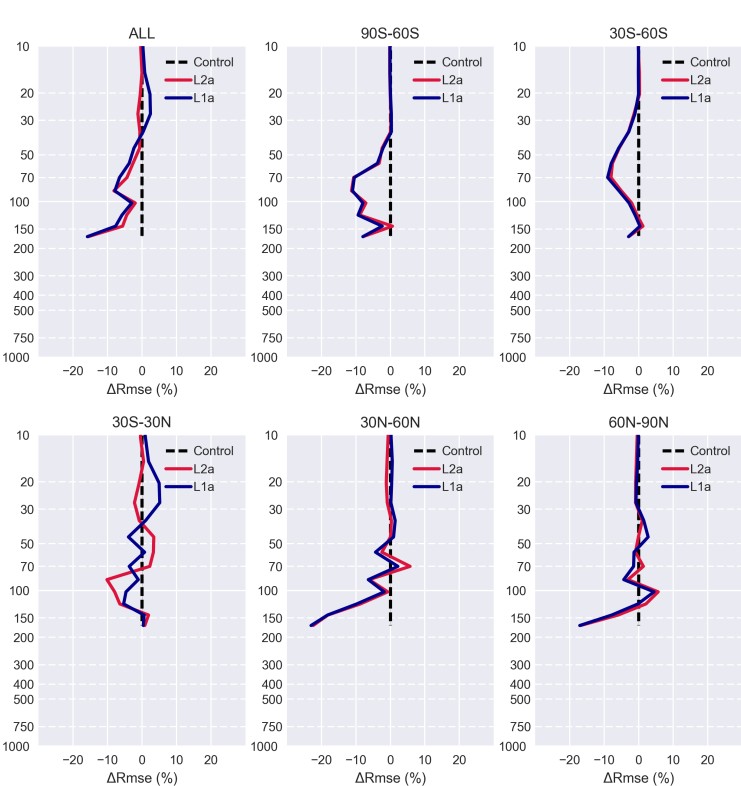

**Figure 4.** Relative difference of RMSE with respect to MLS profiles for L1a (blue) and L2a (red). Same plots as in Fig. 3.





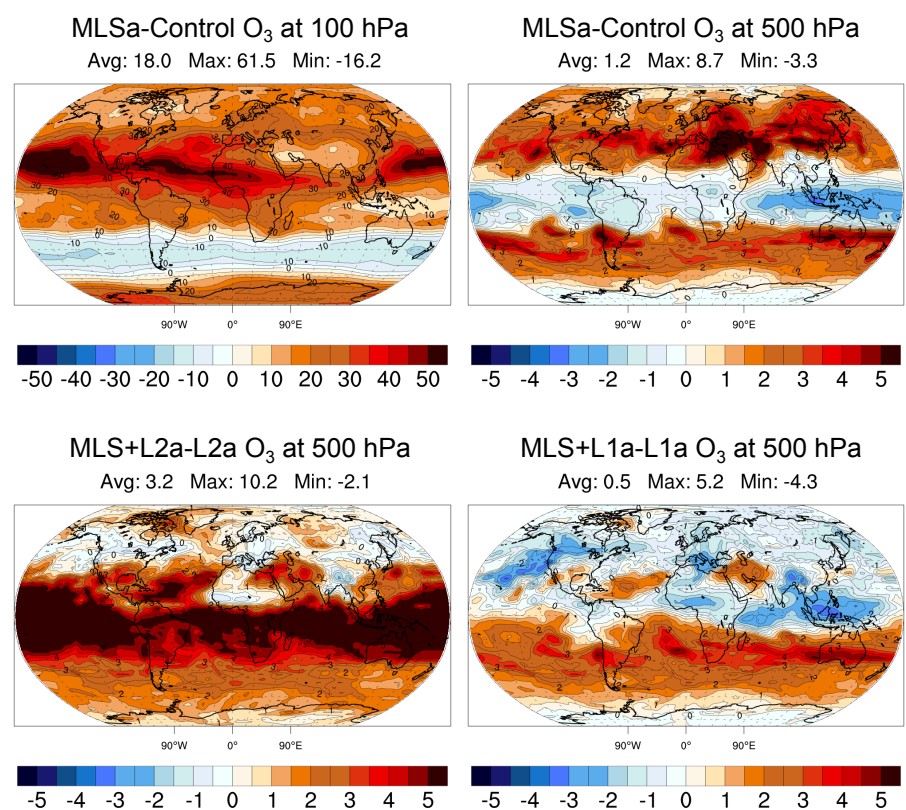

**Figure 5.** Relative differences between $O_3$ reanalyses (as % of the control $O_3$) averaged on July 2010. On top: relative difference between MLSa and the control simulation at 100 hPa (left) and 500 hPa (right). On bottom: relative difference between MLS+L2a and L2a (left) and between MLS+L1a and L1a (right), both at 500 hPa.





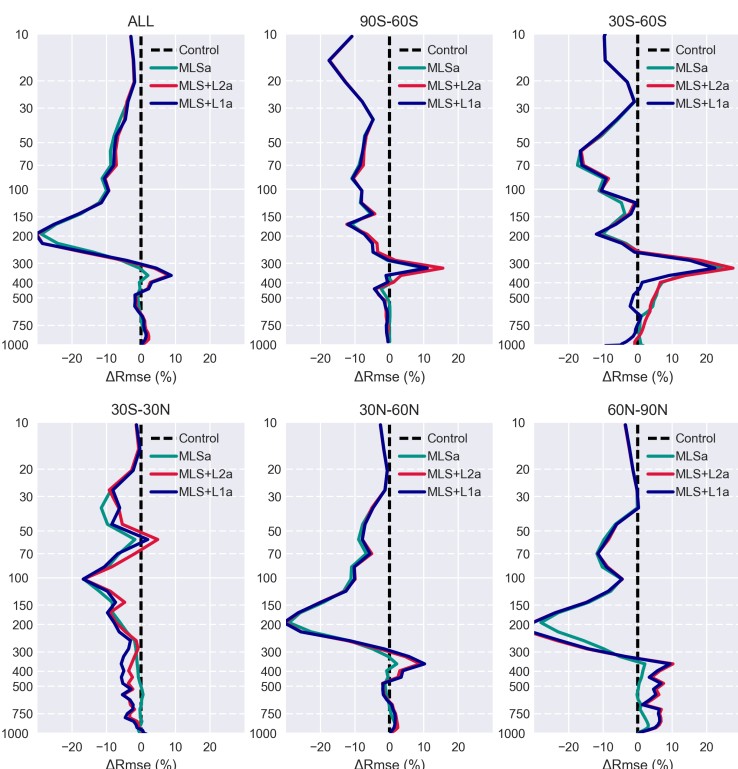

**Figure 6.** Relative difference of RMSE with respect to radiosoundings for MLS-a (teal), MLS+L1a (dark blue) and MLS+L2a (red). Same plots as in Fig. 3.

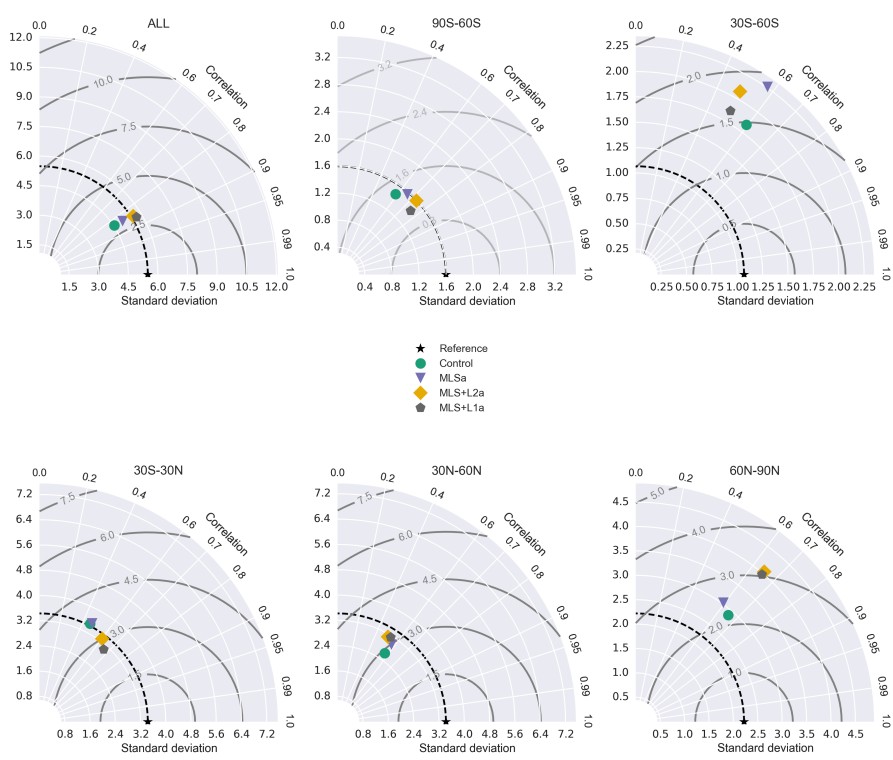

**Figure 7.** Taylor diagrams of modeled tropospheric ozone columns (340-750 hPa) for the Control simulation (green), MLS-a (violet), MLS+L1a (grey) and MLS+L2a (yellow) averaged globally and for five latitude bands separately. The Taylor statistics are computed against radiosoundings.