# Peer review of "Comparison between the assimilation of IASI Level 2 ozone retrievals and Level 1 radiances in a chemical transport model"

_Atmospheric Measurement Techniques, 2018_

## Referee Comment (RC1) · Anonymous Referee #2 · 17 Feb 2019

Main conclusion:

The authors compare the assimilation of IASI ozone L2 retrievals with the assimilation of IASI radiances, with as much as possible the same settings and the same forward RTM. The equivalence between L1 and L2 assimilation depends on e.g. the non-linearity of the relation between radiation and ozone and on the realism of the linearisation point, or a-priori profile used. I agree with the authors that the degree of similarity between L2 and L1 assimilation depends on details and should be studied for real observations. I find this topic very interesting and relevant, and this paper is unique as far as atmospheric trace gas assimilation is concerned. Therefore I am in favour of publishing the results.

However, I have several general and specific comments which require significant adjustments to the present draft. I suggest the responses to these comments are incorporated in a modified manuscript before publishing.

General comments:

The authors document clear differences between L2 and L1 assimilation, and seem to suggest that L1 assimilation should be considered. However, it seems to me that L2 with an improved a-priori may be a possible alternative to reach a similar performance of the analysis. Would running the IASI L2 retrievals with a varying a-priori, for instance taken from the Copernicus Atmosphere Monitoring Service daily analyses, be a feasible option? Would that solve part of the problem with L2 compared to L1? Could the authors discuss this in a more balanced way in the conclusion section (and maybe in the abstract as well)?

Although the L1 and L2 experiments are set up with as much as possible equal inputs and RTM (but different a-priori), there are still subtle differences as discussed in the text. I am wondering how much those differences may also result in differences in performance as documented in the paper? Especially since the differences documented between L1 and L2 are quite small. This leaves me with a bit an uneasy feeling that the results are maybe not fully understood.

The relative differences in Fig. 2 seem to indicate persistent biases. Maybe it is a lot of extra work, but I wonder how the difference plot of L2 retrievals for the climatological a-priori (presented in the paper), compared to L2 retrievals with MOCAGE profiles would look? Such a plot would be a valuable addition. Would that show similar features as in Fig.2, at around 300-500 hPa ?

It would be helpful if the authors could add an image of the IASI averaging kernels, typical examples or averages, for NH, tropics and SH. In this way the reader can better understand at which pressures one may expect an impact of IASI.

The impact of IASI in both the L1 and L2 experiments seems to be relatively small,

with also negative impacts. Especially when MLS is included as well, which already removes most of the bias around 200 hPa. This baseline

Detailed comments:

Title: "for ozone reanalyses": upon first glance this seems to suggest that the paper presents results of a multi-year reanalysis, which is not the case. Is it necessary to include the word "reanalysis" in the title?

Abstract, l9: "significant differences". The abstract does not give a very firm conclusion. Does the work presented justify the stronger statement that the non-linearity in the retrievals in combination with unrealistic a-priori profiles are the cause of the L1-L2 differences?

Abstract: Is there a clear recommendation from this work? Would L1 assimilation be preferred? A more clear statement would be helpful.

page 2

l5: Useful to mention the averaging kernels as well: .. and DOF linked to the averaging kernels ...

l9: "First atmospheric composition models": Reformulate

l19: "However, some aspects of the Data Assimilation (DA) approach differ between the chemistry and meteorology communities." Please be more specific, or remove the sentence.

l21: "Since long time" Please be more specific, by e.g. providing the year when this became common practice.

l23: "This resulted necessary to avoid"; please reformulate.

l24: About the historical background: I was wondering about the problems encountered when assimilating L2 retrievals in NWP (paper of Eyre)? Is it the non-linearity and apriori dependence (as suggested by the text), or is it a simplification of the retrieval results? The latter could arise when e.g. kernels and full covariance matrices are not used in the assimilation, or not provided by the retrieval teams, which would clearly lead to strong a-priori dependence of the analyses.

l27: I would suggest to refer to the book of Rodgers as well. Also the paper of Migliorini 2012 is relevant here.

l29: "Both are based" It is not so clear what "both" refers to: the two studies, or SOFRID and MOCAGE.

page 4

Section 2.1: METOP also has the GOME2 instrument. Has the synergy IASI-GOME2 been considered? Why the choise to use MLS?

page 5

l4: "increased biases": what does "increased" refer to?

l10: "LA" ?

page 7

l11: "ECMWF NWP model" Please replace the word "model" by e.g. "NWP model and assimilation system".

l19: The RTTOV versions for the L1 and L2 experiments are different, see table 1. Can the authors be sure that this does not significantly influence the results/conclusions?

l29: "... and was extended ..."

page 8

l19: "we assimilate here directly the full L2 profiles (43 levels)". Migliorini wrote a paper

(2007) to discuss an efficient interface between L2 retrievals and data assimilation which is relevant in this context. Because the DOF is quite low, this impllies that a lot of noise (43-DOF) is presented to the assimilation when all 43 levels are included. In principle I agree that this avoids any loss of information, but in practice I wonder if the full information may introduce numerical issues (randomness) in the system, especially when this is combined with vertical interpolations? Please comment.

l22: "The steps for the computation of modeled radiances are equal to the profiles ones until the vertical interpolation." Please reformulate.

l27: "climatological profile" -> "climatological profiles"

l29: "as it is done within SOFRID retrieval scheme" please reformulate

l29: Does this mean that the SST is treated differently in the L1 vs L2 assimilation experiments?

page 9

l8: "initialized on 1st June 2010"; replace by "initialized on 1 June 2010"

l11: "a diagonal matrix (i.e. with no inter-channel correlation) is used". Is the same diagonal matrix used in the retrievals that produce the L2 dataset?

p9, bottom to p10, top: I got a bit lost with the numbers provided for the background standard deviation, also in comparison with Fig. 1. I understand that a background standard deviation during assimilation is often smaller than the std of a free model run, but I do not manage to connect the numbers with e.g. Fig.1 in combination with Fig.3?! What is the motivation to go from 5% to 2% in the stratosphere, which seems like a big step and does not seem justified given Fig.1? Does this choice lead to very small stratospheric increments? What is the justification for a step between stratosphere and troposphere? The standard deviation should depend on the data assimilated. Normally these kind of numbers are optimised with e.g. a chi-square test.

p10, l15: One would expect that features in the boundary layer, and, to a lesser extent, the free troposphere show vertical correlations because of e.g. vertical mixing and convection. This in contrast to the stratosphere.

p10, first part: I think the B matrix discussion can be shortened somewhat, because optimising it is not so important for the topic of the paper.

page 11

l5: "are in generally ", remove "in"

l7: "are found in correspondence of tropical latitudes". Please reformulate.

l14: "equivalence between L1 and L2 assimilation is not verified for O3". I suggest to explicitly add "for O3 retrievals in the thermal infrared".

page 12

l2: "The assimilation increases the RMSE of the tropospheric profile at northern latitudes (60ôŘřŞN-90ôŘřŞN)." I guess you mean in the range 350-1000 hPa.

l8: "the other way round". Replace by "around".

l17: "Hence, we expect a stronger impact of the prior in the retrieval results,". I do not understand this. It means that the DOF is smaller, which is clearly observed at altitudes around 200hPa, where the improvement with IASI data is much more limited in the SH. But a-priori plays only a role through non-linear effects. Why would these non-linear effects be larger in the SH?

l27: "The only exceptions are a lower RMSE degradation at 50 hPa". Should we believe the sondes or MLS here? How many sonde launches are included, and confirm the 60 hPa bias?

l32: "total computing time is 3.9 CPU hours". Is this on a single core/node ??

page 13

[Figure]

Fig.5: The figure seems to prove that the analysis of MLS and of IASI are more consistent in the case of L1, while the L2 plot indicates biases between the instruments, especially in the tropics. This could be discussed a bit more explicitly.

page 14

l16: "mixed elsewhere"? Do you mean to say "mixed results are obtained elsewhere"?

l22: "which are differences" please correct the English.

l23: "reanalyses". I suggest to broaden this to e.g. "analyses and reanalyses".

l24: "between the L2 retrieval and the assimilation algorithm ". I suggest to change to " between the L2 retrieval and the L1 observation operator" or something like that

l24: "using the same RTM". But the version of the RTM is different ?!

l28: "between each other". I suggest "against each other".

l30: "Main findings suggest". I suggest "The results suggest ..."

page 15

l6: "We could imagine". I could as well, but is this a recommendation?

l6: The non-linearity of the retrieval may be very different for different species and spectral ranges. Which ones would be candidates to show significant differneces between L1 and L2?

l9: I was wondering if we may expect positive synergies between IASI and GOME2? They are both on the same platform. Please discuss.

l15: "Level 2 products can be aggregated". This useful remark could be phrased more generally by refering to "Use of the Information Content in Satellite Measurements for an Efficient Interface to Data Assimilation" by Migliorini et al, 2007. Through L2 the number of useful observations presented to the assimilation may be optimised (to ultimately match the DOF).

---

## Referee Comment (RC2) · Anonymous Referee #1 · 6 Mar 2019

This article aims at comparing the assimilation of IASI ozone radiances and IASI ozone retrievals in the CTM MOCAGE. I appreciate the authors want to have as few differences between the two settings as possible, as details matter. The subject of the article is interesting and useful for the community. I recommend this paper is published after some (possibly major) modifications. Hereafter are my detailed comment.

General comments: - although both SOFRID retrievals and assimilation of L1 use the RTTOV model, differences may be significant. Indeed the version of the RTTOV coefficients used in both cases is not given. I suspect that SOFRID uses coefficients on 43 levels which are the levels of the retrieval and that the L1 assimilation uses newer coefficients, on 54 or 101 levels (the authors state 104 vertical levels on P7 L20 which does not exist), which may have been build from a different line-by-line model (or dif-

ferent version of that model; or even from a different spectroscopic database). Such differences can have visible impacts on the radiance simulation by RTTOV. Could the authors give mode details about the coefficients? Is it possible to produce SOFRID L2 retrievals using the same version of RTTOV model and RTTOV coefficients as those used in CTM assimilation? And then run the L2 assimilation trials in CTM? That would be a significant improvement to the comparison proposed in this paper!

- ECMWF NWP forecasts are used in both SOFRID and CTM assimilation. In the CTM runs, forecasts are taken from the latest available analysis (00 or 12 UTC) as said in sec. 3.1, supposedly every hour, and scaled on the CTM grid. In the SOFRID retrieval process, are the forecasts from IFS used the same way? Before being fed to RTTOV, the meteorological forecasts have to be interpolated to the location of IASI pixels. I would appreciate that the authors describe how this interpolation is done in SOFRID and in the CTM. ECMWF 4DVAR analyses have ozone in the control variable and assimilate ozone-sensitive information (such as some IASI channels). I would not be surprised that the subsequent ECMWF forecasts are more consistent with the L1 assimilation than with the L2 products. Can the authors elaborate on that point?

- No description of the L1 and L2 innovation statistics is given. Figures on biases and standard deviations of L1 and L2 innovations would be of interest in this paper. How the value chosen for the observation error standard deviation (0.7 mWm-2sr-1) compare to those statistics? Cloud masks are not really described. Cloud fraction from AVHRR is mentioned but no threshold value is given. How clear cases are selected? A data thinning is applied. Which is the minimum distance between two pixels? No description of the spatial coverage of L1 and L2 is given. Would it be possible to have a typical daily coverage or an average density over the month?

- Background covariance error matrix: the values used in this study (2% / 10%) are barely supported by Figure 1. The authors state that the bias may be an important component of the RMSE in Figure 1, is it possible to provide profiles of bias and standard deviation in addition to RMSE? P10 L5, the vertical structure of the B matrix is

described as "correlation length of 1 model grid point". Do you mean 1 model level? Please clarify.

- Results L1 vs L2 Figure 2 shows the relative differences between L1 analyses and L2 analyses. As the values of background error variances are rather small, I would find interesting to show analysis increments difference statistics (average and/or standard deviation). All figures are given in relative difference and no ozone fields are plotted. Except from the value given P11 L6, the reader have no idea how these relative differences compare to the actual ozone concentration. I would appreciate the authors find a way to illustrate the 3D field they want to analyse in their study. Figure 3 (and similar figures) would be more useful if error bars were added. They would help understand whether the differences are statistically significant or not. The statistics are given over the whole month. How stable are they on a day to day basis? Would it be interesting to split the statistics between day and night? The paragraph about the computational cost and convergence issues is interesting but may be placed separately from the scientific results.

- Results when MLS is assimilated As in a real system, several sources of observation may be assimilated simultaneously, this section has a real added value. I regret that the results are not shown in a consistent way with the previous section. Figure 2 shows L1a - L2a; Figure 5 should show MLS+L1a - MLS+L2a because we want to compare these two settings.

Minor: - P2 L26: please detail what Averaging Kernels are for non-expert readers. - P2 L27: the innovation has not been described so far, please define innovation. - P4 L8: the IASI acronym is already detailed P3 L18 - P4 L11: please add Hilton et al (BAMS 2012) reference to Clerbaux et al (2009) - P4 L13: the authors state "A total of 3 IASI ... providing nearly global coverage 3 times per day". Each IASI has a nearly global coverage twice a day (morning and evening overpasses) thus the global coverage will be achieved 6 times a day. - P4 L16: IASI exactly has 8461 channels per spectrum - throughout the paper: MetOP should be written Metop according to Eumetsat

---

## Author Comment (AC1) · 18 May 2019

**Comparison between the assimilation of IASI Level 2 retrievals and Level 1 radiances for ozone reanalyses. Reply to referee # 1**

Emanuele Emili[1], Brice Barret[2], Eric Le Flochmoën[2], and Daniel Cariolle[1]

[1]CECI, Université de Toulouse, Cerfacs, CNRS, Toulouse, France
[2]Laboratoire d'Aérologie, Université de Toulouse, CNRS, UPS, Toulouse, France

**Correspondence:** Emili (emili@cerfacs.fr)

**1 Reply to general comments**

We thank the anonymous reviewer for his comments, which helped to improve significantly the manuscript. Detailed replies to his comments follow:

1. *Although both SOFRID retrievals and assimilation of L1 use the RTTOV model, differences may be significant. Indeed the version of the RTTOV coefficients used in both cases is not given. I suspect that SOFRID uses coefficients on 43 levels which are the levels of the retrieval and that the L1 assimilation uses newer coefficients, on 54 or 101 levels (the authors state 104 vertical levels on P7 L20 which does not exist), which may have been build from a different line-by-line model (or different version of that model; or even from a different spectroscopic database). Such differences can have visible impacts on the radiance simulation by RTTOV. Could the authors give mode details about the coefficients? Is it possible to produce SOFRID L2 retrievals using the same version of RTTOV model and RTTOV coefficients as those used in CTM assimilation? And then run the L2 assimilation trials in CTM? That would be a significant improvement to the comparison proposed in this paper!.*

   **Answer**:

   A verification of the differences between the versions of RTTOV used for this study (v9 for SOFRID and v11.3 for L1 assimilation) confirmed the concerns of the reviewer: SOFRID retrievals were based on a mixture of HITRAN 2000 and 2004 spectroscopic databases, LBLRTM v11.1 radiative transfer and predictors computed for 43 levels, whereas L1 assimilation uses HITRAN 2008, LBLRTM v12.2 and predictors on 101 vertical levels. Therefore, we switched to an updated version of SOFRID and recomputed L2 retrievals with RTTOV 11.1 and the same predictors used for L1 assimilation (101 levels). Other minor differences between RTTOV 11.1 and 11.3 do not impact the radiance computations. New L2 retrievals using RTTOV 11 are named v3.0, as opposed to v1.6 used for the original manuscript. A short summary of the different versions of SOFRID used for this study is given in Tab. 1. The further assimilation of v3.0 retrievals confirmed the results observed previously at tropical latitudes but reduced significantly the differences between L1a and L2a in the Southern Hemisphere mid-latitudes (Fig. 1 and 2). We conclude that the interpretation of the results in the

[Figure]

**Figure 1.** Relative differences (%) between radiances and Level 2 assimilation (L1a minus L2a divided by the correspondent $O_3$ values of the control simulation) averaged on July 2010. From left to right different pressure levels are displayed covering the stratosphere (top) and the free troposphere (bottom). Average, maximum and minimum values of the displayed fields are given on top of each map. The same pixels selection of the original manuscript is used to produce this figure. The only difference with the original manuscript is that SOFRID v3.0 retrievals are used instead of v1.6 (Tab. 1)

SH mid-latitudes given in the original manuscript was not correct: the L1 assimilation does not reduce the biases when the instrument's sensitivity is low thanks to a better prior. The better performances of L1a in the SH mid-latitudes were mostly due to improved radiative transfer computations.

These findings made us revise the discussion of the results (Sec. 4.1 and 4.3 of the revised manuscript) and, partly, the conclusions. Even if the positive results of the original manuscript are somehow mitigated in the SH, the main conclusions remain valid elsewhere. Moreover, we can now provide a more satisfactory explanation of the differences between L1a and L2a: large differences arise only where the model departures from the SOFRID prior are very large (> 100%), i.e. at low latitudes ($< 40°$). As the second referee also pointed out, differences between L1a and L2a do not really depend on the sensitivity of the instrument but on the accuracy of the prior and on the consequent linearization of the RT. After switching to the same version of RTTOV the results better support this explanation. Please refer to the replies n. 2 and 32 to the second referee for a more detailed discussion on this point.

2. *ECMWF NWP forecasts are used in both SOFRID and CTM assimilation. In the CTM runs, forecasts are taken from the latest available analysis (00 or 12 UTC) as said in sec. 3.1, supposedly every hour, and scaled on the CTM grid. In the SOFRID retrieval process, are the forecasts from IFS used the same way? Before being fed to RTTOV, the meteorological forecasts have to be interpolated to the location of IASI pixels. I would appreciate that the authors describe how this*

[Figure]

**Figure 2.** Relative difference of RMSE (ΔRMSE) with respect to radiosoundings for L1a (blue) and L2a (red). The difference is computed by subtracting the RMSE of L1a (L2a) against radiosoundings from the RMSE of the control simulation. Negative values mean that the assimilation improved (decreased) the RMSE of the control simulation, positive values indicate degradation (increase) of the RMSE. The same pixels selection of the original manuscript is used to produce this figure. The only difference with the original manuscript is that SOFRID v3.0 retrievals are used instead of v1.6 (Tab. 1).

**Table 1.** Versions of SOFRID used for the original and revised manuscript.

| SOFRID Version | T and H2O | Cloud factor | RTTOV version | Usage |
|---|---|---|---|---|
| 1.5 (Barret et al., 2011) | EUMETSAT L2 | EUMETSAT plus L1 | 9.0 | Original manuscript (only cloud factor) |
| 1.6 | ECMWF NWP | L1 | 9.0 | Original manuscript (except cloud factor) |
| 3.0 | ECMWF NWP | EUMETSAT plus L1 | 11.1 | Revised manuscript |

*interpolation is done in SOFRID and in the CTM. ECMWF 4DVAR analyses have ozone in the control variable and assimilate ozone-sensitive information (such as some IASI channels). I would not be surprised that the subsequent ECMWF forecasts are more consistent with the L1 assimilation than with the L2 products. Can the authors elaborate on that point?*

**Answer**:

The meteorological forcing of MOCAGE is retrieved from the ECMWF MARS servers with a 3 hours stepping (available steps for the forecast type "fc"), further regridded to the CTM resolution ($2°x2°$) and stored as input files. During the MOCAGE execution the meteorological fields are interpolated linearly at hourly sub-steps, which corresponds to the advection time step of the CTM, and vertically (91 to 60 levels, linear interpolation). The observation operator performs an additional bi-linear interpolation at the position of each observation and a linear interpolation to the observation's time. The obtained profiles (temperature, water vapor and ozone on the CTM levels) are used to feed RTTOV, which is in charge of the final vertical interpolation to the coefficients levels. As a consequence, both the spatial and temporal resolution of the RTM vertical profiles are degraded with respect to the original NWP forecasts but are coherent with the resolution of the CTM model. On the other hand, surface properties such as surface skin temperature, which are only needed for the RT and might display a larger variability at smaller scales than the CTM resolution are taken from higher resolution IFS fields ($0.125°x0.125°$) and interpolated at the IASI pixel using nearest neighbor approach.

SOFRID preprocessor retrieves the IFS operational analysis (type "an") at 00-06-12-18 UTC, regridded to a resolution of $0.25°$ x $0.25°$. All the fields are then interpolated at the closest hour to the IASI pixel and a nearest neighbor interpolation is done to extract the corresponding profiles and surface properties. These information have been added to the revised manuscript (Sec. 2.1.2 and 3.1) and Table 1 (now Table 2 in the revised manuscript) was upgraded accordingly.

Hence, differences between L1a and L2a due to the different origin, resolution and interpolation of the temperature and water vapor profiles might contribute to differences observed in our results. However, we assimilated the IASI main ozone window in the study (980-1100 cm$^{-1}$) and channels with strong sensitivity to water vapor were excluded both in L1a and L2a (Sec. 2.4). Therefore, we expect the impact of the meteorological profiles on our results to be minor. To confirm this we rerun all the experiments of the manuscript using ERA interim instead of the operational NWP forecasts to force the CTM. ERA interim not only differs in the model configuration with respect to the NWP operational model (e.g. 60 vertical levels for ERA interim versus 91 for the NWP model in 2010), but also for the assimilation: for example no IASI data are assimilated within ERA interim. This introduces some differences in the RTM computations for L1a but also in the control O$_3$ fields through the CTM forcing, thus requiring to recompute L1a, L2a and the control simulation. Since L2 products are kept the same, potential differences between L1a and L2a due to the meteorological profiles are now amplified. We show in Fig. 3 the same plots as in Fig. 6 (and revised manuscript) but computed using ERA interim forcing. The differences between L1a and L2a at the tropics show similar patterns to previous results and suggest that the main results of this study are not a consequence of the different meteorological profiles. We kept the original choice for the meteorological profiles in the revised manuscript and added a sentence to discuss this point (page 14, line 20).

[Figure]

**Figure 3.** Relative differences (%) between radiances and Level 2 assimilation (same plots as in Fig. 6) but forcing the CTM with ERA-interim analyses instead of ECMWF operational forecasts.

The surface skin temperature has a strong signature in the IASI $O_3$ window and, even if it is included in the control vector, the different background values used in SOFRID and L1a might have an impact on our results. Hence, we replaced the surface skin temperature used originally in L1a (IFS 3-hourly forecasts at $0.125°$x$0.125°$) with the one already used in SOFRID (IFS 6-hourly analysis at $0.25°$x$0.25°$). Results in Fig. 4 show that the choice of the background skin temperature has not a significant impact on our results. We kept the original choice in the revised manuscript and added a sentence to discuss this point (page 14, line 18).

We do not fully understand the referee's comment about the "better consistency of L1 assimilation with ECMWF forecasts than L2 products": even if some IASI ozone channels are assimilated in IFS, we do not make any use of ozone fields from NWP forecasts in our study. Hence, we do not expect any particular advantage for L1 assimilation compared to L2 assimilation due to the meteorological forcing itself. Conversely, the fact that SOFRID (v1.6 and 3.0) uses the IFS analyses should in principle make SOFRID background radiances closer to L1 observations than the CTM ones, which uses instead forecast fields (also at a degraded spatial resolution).

3. *No description of the L1 and L2 innovation statistics is given. Figures on biases and standard deviations of L1 and L2 innovations would be of interest in this paper. How the value chosen for the observation error standard deviation (0.7 mWm-2sr-1) compare to those statistics? Cloud masks are not really described. Cloud fraction from AVHRR is mentioned but no threshold value is given. How clear cases are selected? A data thinning is applied. Which is the minimum distance between two pixels? No description of the spatial coverage of L1 and L2 is given. Would it be possible to have a typical daily coverage or an average density over the month?*

[Figure]

**Figure 4.** Relative differences (%) between radiances and Level 2 assimilation (same plots as in Fig. 6) but using exactly the same background surface skin temperature for L1 assimilation and L2 retrievals.

**Answer**:

The answer to this question is split in two parts.

**Innovation statistics**: The innovation statistics represent one of the main diagnostics of data assimilation experiments and have been carefully evaluated during the study. We report for example in Fig. 5 the average innovations of L1 assimilation (experiment L1a) for the entire month of July 2010. We remark for example that the biases of the control simulation are moderate (about 1 $mWm^{-2}sr^{-1}cm$) and that the background (forecast) innovation in the middle of the spectral window is smaller than on the tails. The latter is likely due to the different spectral contributions of ozone and skin temperature and the fact that the skin temperature is not a prognostic variable of the CTM (i.e. the background SST is the same in the control and in the forecast). The value of the observation error was deliberately fixed equal to the one used for L2 retrievals to compare L1a and L2a for same settings. Improvements of the observation error covariance, potentially with the aid of more detailed innovation analysis, are left for a future study (page 15, line 22 of the original manuscript). Even though this type of plots contain highly valued information, we prefer not including them since they are not essential for the conclusions of this study and to avoid an excessive length of the manuscript. Moreover, L1 and L2 innovations are not directly comparable because of their different nature (radiances and profiles).

**Preprocessing method**: the results in Fig. 1 and 2 of this document have been obtained assimilating exactly the same satellite pixels as in the original manuscript, and were shown here to highlight differences due only to the RTTOV version. These pixels were selected based on a combination of cloud masks from an older version of SOFRID (v1.5, see Tab. 1 and description in Sec. 2.1, page 5, line 15) and AVHRR cloud mask available only in most recent L1c files.

[Figure]

**Figure 5.** Average (left) and standard deviation (right) of the L1 innovations for the entire period of simulation. The control line represents the innovation with respect to the control simulation (no DA). The dashed turquoise line represents the observation error standard deviation used for the SOFRID retrievals and L1a experiments.

Due to the different data processors these cloud masks are not the same. For both data sources only pixels with a cloud factor smaller than 1% were first selected. The information was given at page 6, line 18 of the original manuscript. The resulting datasets were colocated to ensure that a valid SOFRID retrieval was available for each L1c pixels. Finally, a data thinning was performed hourly: we covered the Earth with a 1°x1° grid and within an hourly loop we retained only the first satellite pixel found within every two grid boxes. A minimum distance of 1° before assimilated pixels is therefore ensured. Overall, the selection resulted in about 3300 pixels per day for the assimilation, as mentioned in Sec. 2.4 of the original manuscript.

With version v1.6 (the retrievals assimilated in the original manuscript), SOFRID was upgraded to use water vapor and temperature profiles from IFS instead of EUMETSAT L2 retrievals (Tab. 1). This increased the number of retrieved pixels with respect to v1.5, since SOFRID was not subject anymore to the availability of the EUMETSAT Level 2. On the other hand, the original cloud mask of v1.5 based on both L1 spectra and EUMETSAT processor was replaced by the L1-only based mask (described in Sec 2.2 of Barret et al. (2011)). To avoid possible cloud contamination the best option was then to keep the original pixel selection done initially with SOFRID v1.5 but using the retrievals from v1.6. Therefore, all results presented in the original manuscript were based on about 3300 assimilated observations per day (page 6, line 22).

With SOFRID v3.0 (RTTOV 11) the EUMETSAT cloud mask was reintroduced in the L2 product, and allowed to apply the full preprocessing procedure described above but using only v3.0 files. At the end, this resulted in an increased number of pixels available for each day to about 5000 (Fig. 8). Differences between L1 and L2 assimilation are enhanced due to the higher number of assimilated observations (Fig. 6 and 7), but show the same patterns as in Fig. 1 and 2. We retained this configuration for the revised manuscript, we extended the description of data thinning (page 7, line 25) and we included a new plot showing the number of assimilated observations per grid point during the simulation period (Fig.

[Figure]

**Figure 6.** Relative differences (%) between radiances and Level 2 assimilation (same plots as in Fig. 1) but using the new set of colocated observations (right plot in Fig. 8) and SOFRID V3.0 (RTTOV 11). This figure replaces Fig. 2 of the original manuscript.

8, right plot). All figures in this document (except Fig. 1 and 2) and in the revised manuscript are based on this new set of experiments with increased number of observations.

4. *Background covariance error matrix: the values used in this study (2% / 10%) are barely supported by Figure 1. The authors state that the bias may be an important component of the RMSE in Figure 1, is it possible to provide profiles of bias and standard deviation in addition to RMSE? P10 L5, the vertical structure of the B matrix is described as "correlation length of 1 model grid point". Do you mean 1 model level? Please clarify.*

**Answer**:

We extended Figure 1 with the full validation statistics of the control simulation (bias, standard deviation and RMSE), which are reported here in Fig. 9,10 and 11 respectively. The validation values obtained against MLS are also added to these plots for completeness. We remark that biases can be as high as 30% close to the tropopause and that standard deviation and RMSE values relative to MLS are generally smaller due to the increased accuracy and number of MLS observations. Even if we consider MLS lines as reference for the stratosphere, the values chosen for the background standard deviation may still seem small with respect to those in Fig. 10. However, the assimilation background is more accurate than the control simulation (Fig. 12 for the MLSa forecast), with RMSE values that fell generally below 5% in the stratosphere. We also remind that we neglected the radiances error correlation in our study. This leads to a stronger weight of the assimilated observations, that we compensated by smaller values of the background error covariance. Values of 5%-25% for the standard deviation in the stratosphere and troposphere respectively lead typically to worse

[Figure]

**Figure 7.** Relative difference of RMSE (ΔRMSE) with respect to radiosoundings for L1a (blue) and L2a (red) (same plots as in Fig. 2) but using the new set of colocated observations (right plot in Fig. 8) and SOFRID V3.0 (RTTOV 11). This figure replaces Fig. 3 of the original manuscript.

reanalyses (not shown). Using a relatively small error in the stratosphere (2%) mitigated the issues encountered with IASI assimilation (L1 and L2) and did not reduce significantly the positive impact of MLS.

This study is focused on comparing L1 and L2 assimilation with identical values for B: we think that the empirical choices for the B matrix are satisfactory for the objective of the study. Further optimization of B and R, which is often done simultaneously (Desroziers et al., 2005), is left for a future study, where non-diagonal terms of R should also be included in L1 assimilation. We extended the discussion of the background error covariance to include elements from this reply and the reply n 24 to the second reviewer (page 12, lines 6 and 12). We also precised that the scale of the vertical error correlation is expressed in number of model levels (page 13, line 1).

[Figure]

**Figure 8.** Number of assimilated observations for each model grid point ($2°x2°$) and for the entire simulation period (July 2010). Number of observations used in the original manuscript on the left (SOFRID 1.6) and for the revised manuscript on the right (SOFRID 3.0). The total number of observations is displayed on the top of each plot.

5. *Results L1 vs L2 Figure 2 shows the relative differences between L1 analyses and L2 analyses. As the values of background error variances are rather small, I would find interesting to show analysis increments difference statistics (average and/or standard deviation). All figures are given in relative difference and no ozone fields are plotted. Except from the value given P11 L6, the reader have no idea how these relative differences compare to the actual ozone concentration. I would appreciate the authors find a way to illustrate the 3D field they want to analyze in their study. Figure 3 (and similar figures) would be more useful if error bars were added. They would help understand whether the differences are statistically significant or not. The statistics are given over the whole month. How stable are they on a day to day basis? Would it be interesting to split the statistics between day and night? The paragraph about the computational cost and convergence issues is interesting but may be placed separately from the scientific results.*

**Answer**:

The answer is split in 3 parts.

Increments, like innovations, are the direct output of the variational minimization and are among the first diagnostics that we looked at. Examples of increments for the third assimilation window (2010-07-01 03 UTC) are shown in Fig. 13 and 14, which confirms that the absolute increment values are significant in term of typical ozone concentrations (Fig. 15). However, while this type of plots is very meaningful to verify the correct functioning of the DA system, we found not relevant to report average increments in the manuscript. With hourly DA windows the increments are equal to zero most of the time on the global grid due to the moving observation network. Hence, averaged increments do not give valuable information in terms of absolute or relative values. Weighting the average based on the satellite overpasses is not straightforward. On the other hand, the cumulative effect of all increments during the evaluation period is well represented by the analysis fields, which is also the only field that we validated against independent measurements. We think that presenting only the analysis statistics is the best choice for the objective of our study and to avoid an excessive length of the manuscript.

[Figure]

**Figure 9.** Relative bias of the control simulation with respect to radiosoundings (solid line) and MLS (dotted line) averaged globally (first plot) and for five latitude bands separately (90°S-60°S, 60°S-30°S, 30°S-30°N, 30°N-60°N, 60°N-90°N).

We chose to display only relative differences and relative improvements because ozone varies on an exponential scale. When showing absolute values it is often difficult to appreciate the impact of data assimilation on both the troposphere and the stratosphere, especially when examining differences between similar assimilation experiments. We report in Fig. 15 the average value of ozone of the control simulation, which are used to scale all the maps presented in the study. This figure has been included and commented in the revised manuscript (Sec. 4.1).

Figure 3, 4 and 6 of the original manuscript (and Fig. 2,7 in this document) represent differences of RMSE between the analyses and the control simulation. The RMSE for each simulation (e.g. Fig. 11 for the control simulation) is based on the differences between modeled and observed values for the ensemble of the observations, or for a selection based on latitude. It is not clear to us how to put error bars on such statistics. The statistical significance depends on the number of observations used to compute the various RMSEs, which are reported now in Table 2 and included in the

[Figure]

**Figure 10.** Relative standard deviation of the control simulation. Same plots as in Fig. 9.

revised manuscript for completeness. By looking at observation numbers we recognize that daily RMSE statistics would be difficult to compute for radiosoundings due to a too small number of observations. Similar issues arise if we try to separate between day and night, since radiosoundings are mostly launched at local noon. On the other hand, MLS allows to compute daily or night/day statistics.

5    We present in Fig. 16 the same plots as in Fig. 4 of the original manuscript but for five different days during the simulation period. We remark that the RMSE display a tendency during the period, with a slow degradation towards the end of the period. We suppose that, without MLS joint assimilation, some errors are continuously injected by IASI, especially in the case of L1a. This points to some unresolved issues with the inversion of radiances, which is probably exacerbated in L1a because of the propagation of the $O_3$ prior in time. An evaluation of the results over a longer period seem necessary

10   to draw more robust conclusions on this issue. However, thanks to the MLS assimilation, this issue has a limited impact on the main results of the study.

[Figure]

**Figure 11.** Relative Root Mean Square Error (RMSE) of the control simulation. Same plots as in Fig. 9.

Fig. 17 reports the RMSE statistics for the full period but split between day and night. The day-night separation is computed based on the local sun position at the time of the observation. We remark that some differences appear only at high latitudes (90°S-60°S and 60°N-90°N), but since the number of day-night observations changes dramatically in these regions (e.g. from 15755 to 1212 at 90°S-60°S), a robust interpretation of these differences looks problematic.

The paragraph on the computational cost has become an independent section in the revised manuscript (Sec. 4.2).

6. *Results when MLS is assimilated As in a real system, several sources of observation may be assimilated simultaneously, this section has a real added value. I regret that the results are not shown in a consistent way with the previous section. Figure 2 shows L1a - L2a; Figure 5 should show MLS+L1a - MLS+L2a because we want to compare these two settings.*

**Answer**:

We agree with the reviewer and the previous figure has been replaced in the revised manuscript with MLS+L1a - MLS+L2a (Fig. 18). The new plot shows that differences are largely reduced in the stratosphere, thanks to MLS, but

[Figure]

**Figure 12.** Relative Root Mean Square Error (RMSE) of the control simulation (black lines) and of the MLSa forecast (green lines). Same plots as in Fig. 11.

**Table 2.** Number of validation observations.

| Latitudes | MLS | Radiosoundings |
|-----------|-----|----------------|
| Global | 100975 | 219 |
| 90°S-60°S | 16967 | 19 |
| 60°S-30°S | 17334 | 9 |
| 30°S-30°N | 33046 | 38 |
| 30°N-60°N | 16669 | 138 |
| 60°N-90°N | 16959 | 15 |

[Figure]

**Figure 13.** Absolute O$_3$ increments (ppb units) in L1a experiment for the 2010-07-01 03 UTC window and at different pressure levels in the stratosphere (top plots) and in the free troposphere (bottom plots).

[Figure]

**Figure 14.** Absolute O$_3$ increments (ppb units) in L2a experiment for the 2010-07-01 03 UTC window and at different pressure levels in the stratosphere (top plots) and in the free troposphere (bottom plots).

are still significant in the free troposphere, although to a lesser extent than for L1a-L2a (Fig. 6). As a consequence, the discussion of the previous figure has also been removed from the revised manuscript.

[Figure]

**Figure 15.** O$_3$ fields (ppb units) issued from the control simulation averaged on July 2010. From left to right different pressure levels are displayed covering the stratosphere (top) and the free troposphere (bottom). Average, maximum and minimum values of the displayed fields are given on top of each map.

[Figure]

**Figure 16.** Gain of RMSE (ΔRMSE) computed with respect to MLS for L1a (blue) and L2a (red). Same plots as in Fig. 4 of the original manuscript but shown only for the global average and for five different dates.

**2   Reply to specific comments**

**Answer**:

All specific comments have been integrated in the revised manuscript.

Day                             Night

[Figure]

**Figure 17.** Gain of RMSE (ΔRMSE) computed with respect to MLS for L1a (blue) and L2a (red). Same plots as in Fig. 4 of the original manuscript but computed using only observations during daylight (left panel) and night (right panel).

**References**

Barret, B., Le Flochmoen, E., Sauvage, B., Pavelin, E., Matricardi, M., and Cammas, J. P.: The detection of post-monsoon tropospheric ozone variability over south Asia using IASI data, Atmospheric Chemistry and Physics, 11, 9533–9548, https://doi.org/10.5194/acp-11-9533-2011, http://www.atmos-chem-phys.net/11/9533/2011/, 2011.

5   Desroziers, G., Berre, L., Chapnik, B., and Poli, P.: Diagnosis of observation, background and analysis-error statistics in observation space, Quarterly Journal of the Royal Meteorological Society, 131, 3385–3396, https://doi.org/10.1256/qj.05.108, http://doi.wiley.com/10.1256/qj.05.108, 2005.

[Figure]

**Figure 18.** Relative differences (%) between L1a+MLS minus L2a+MLS divided by the correspondent O$_3$ values of the control simulation averaged on July 2010. From left to right different pressure levels are displayed covering the stratosphere (top) and the free troposphere (bottom). Average, maximum and minimum values of the displayed fields are given on top of each map. This figure replaces Fig. 5 of the original manuscript.

---

## Author Comment (AC2) · 18 May 2019

**Comparison between the assimilation of IASI Level 2 retrievals and Level 1 radiances for ozone reanalyses. Reply to referee # 2**

Emanuele Emili[1], Brice Barret[2], Eric Le Flochmoën[2], and Daniel Cariolle[1]

[1]CECI, Université de Toulouse, Cerfacs, CNRS, Toulouse, France
[2]Laboratoire d'Aérologie, Université de Toulouse, CNRS, UPS, Toulouse, France

**Correspondence:** Emili (emili@cerfacs.fr)

**1   Reply to general comments**

We thank the anonymous reviewer for his comments that helped to improve significantly the original manuscript. Detailed replies to his comments follow:

1. *The authors document clear differences between L2 and L1 assimilation, and seem to suggest that L1 assimilation should be considered. However, it seems to me that L2 with an improved a-priori may be a possible alternative to reach a similar performance of the analysis. Would running the IASI L2 retrievals with a varying a-priori, for instance taken from the Copernicus Atmosphere Monitoring Service daily analyses, be a feasilble option? Would that solve part of the problem with L2 compared to L1? Could the authors discuss this in a more balanced way in the conclusion section (and maybe in the abstract as well)?*

   **Answer**:

   The reviewer is right about the fact that L2 $O_3$ retrievals can be improved through a better a-priori (see reply n 3 for more details), and using $O_3$ forecast fields from Copernicus Services might represent a particularly valuable option for L2 production itself. We think that such option could reduce the differences between L1a and L2a in our experiments. However, more generally, the same question raised by our study concerns also models within the Copernicus services themselves (e.g. C-IFS). To assimilate L2 $O_3$ profiles in C-IFS we would then need to: i) run a first analysis/forecast excluding all IASI $O_3$ channels ii) run the L2 processor iii) run a second analysis/forecast cycle including only IASI L2 retrievals. On top of the extra numerical cost of such a system and practical difficulties when many instruments are assimilated, this could introduce error correlations between assimilated observations and model forecast that are not yet considered in DA algorithms. Another issue arises for spectral channels that are sensitive to multiple model variables (e.g. T and $O_3$): splitting the DA problem in different steps (e.g. 4D-Var for T plus 1D-Var for $O_3$) would result in the same observation to be used twice and might lead to different solution than solving the full problem at once (4D-Var). We have not investigated these aspects in our study and we cannot give final words on the best choice between L1 and L2 assimilation based only on our results in such a context. However, the ensemble of our results plus all the previous

arguments suggest that the L1 assimilation should deserve higher consideration, especially in the context of coupled systems such as the Copernicus Monitoring Services. Therefore, in general we prefer not to suggest an upgrade of the L2 processor with modeled a-priori for the scope of further assimilation. We included these elements in the revised conclusions (page 19, line 11).

2. *Although the L1 and L2 experiments are set up with as much as possible equal inputs and RTM (but different a-priori), there are still subtle differences as discussed in the text. I am wondering how much those differences may also result in differences in performance as documented in the paper? Especially since the differences documented between L1 and L2 are quite small. This leaves me with a bit an uneasy feeling that the results are maybe not fully understood.*

**Answer**:

Following the comments of the 1st referee we repeated all the experiments using exactly the same version of the radiative transfer for L1a and L2 retrievals (RTTOV 11, see reply n 1 to the 1st referee). We also verified if the differences on the meteorological profiles and surface skin temperature between L1a and L2 retrievals did not impact our results (reply n 2 to the 1st referee). This reduced further the possible sources of differences between the two approaches, which are now limited to: the a-priori, the vertical resolution and the minimization (3D-Var for L1a versus 1D-Var+3D-Var for L2a). Although the L1a-L2a differences are now much reduced in the SH mid-latitudes (original differences were due to the radiative transfer), the new results still show significant differences between L1a and L2a at low latitudes (as high as 30%). We report in Fig. 1 the average difference between the control simulation and the SOFRID a-priori, which show that departures are very large (> 50% and as high as 700%) only at low latitudes. Differences in percent are very large close to the tropical tropapause (150 hPa) because the SOFRID a-priori is representative of mid-latitudes. However, we remark that differences larger than 100% exist also in the free troposphere (300 to 750 hPa). Hence, Fig. 1 confirms that differences within data assimilation arise only when the L2 a-priori is strongly biased (i.e. at low latitudes, see also reply n 32) and strengthens the interpretation of the results given in the original manuscript. We included Fig. 1 in the discussion section (Sec. 4.1) and updated the conclusions of the revised manuscript (page 19, lines 5-8). We think that with these new elements the interpretation of the results is now more robust.

3. *The relative differences in Fig. 2 seem to indicate persistent biases. Maybe it is a lot of extra work, but I wonder how the difference plot of L2 retrievals for the climatological a-priori (presented in the paper), compared to L2 retrievals with MOCAGE profiles would look? Such a plot would be a valuable addition. Would that show similar features as in Fig.2, at around 300-500 hPa ?*

**Answer**:

An evaluation of SOFRID retrievals using an a-priori issued from a model was performed prior to this study and was actually the main motivation for this work. Indeed, results showed significant differences in the L2 tropospheric columns with the modeled a-priori and a better agreement with independent data (Fig. 2). Differences between the two SOFRID datasets seem qualitatively coherent with the L1a-L2a plot in the revised manuscript (increased $O_3$ at 300-500-750

[Figure]

**Figure 1.** Relative differences (%) between control simulation and SOFRID a-priori (divided by the correspondent $O_3$ values of the control simulation) averaged on July 2010. From left to right different pressure levels are displayed covering the stratosphere (top) and the free troposphere (bottom). Average, maximum and minimum values of the displayed fields are given on top of each map.

hPa). However, these experiments were based on a different model configuration (linearized chemistry) with degraded resolution ($10°$x$20°$). This preliminary analysis was also limited to tropical latitudes and integrated $O_3$ columns were evaluated without considering averaging kernels. Differently from our manuscript, the above analysis is focused on the L2 retrievals themselves (without further assimilation) and is still under finalization (comparison with radiosoundings). It will be presented in a separate paper once it is finalized and we prefer not to include partial results in our manuscript. In particular, the analysis of averaging kernels cannot be neglected when the assimilation is concerned, which limits the interest of Fig. 2 for our manuscript. Even so, we added a sentence in the conclusion (page 19, lines 12-13) to link these preliminary results to our study.

4. *It would be helpful if the authors could add an image of the IASI averaging kernels, typical examples or averages, for NH, tropics and SH. In this way the reader can better understand at which pressures one may expect an impact of IASI.*

    **Answer**:

    We report in Fig. 3 the average kernels of the SOFRID retrievals for the month of July 2010, averaged globally and by latitude band. We included this figure in the revised manuscript and added the relative discussion (Sec. 2.1.2).

5. *The impact of IASI in both the L1 and L2 experiments seems to be relatively small, with also negative impacts. Especially when MLS is included as well, which already removes most of the bias around 200 hPa. This baseline ...*

    **Answer**:

[Figure]

**Figure 2.** Average $O_3$ tropospheric columns (1000-100 hPa) from OMI-MLS residual method of Ziemke et al. (2011) (top), SOFRID v1.5 standard retrievals (middle) and SOFRID retrievals issued from a modeled a-priori (bottom).

The question of the reviewer being incomplete, we suppose that he/she raises some doubts about the practical benefits of assimilating IASI on top of MLS for $O_3$. MLS assimilation is able to well correct the upper troposphere $O_3$ but, as Fig. 4 and 5 show, there is a significant positive correction of IASI in the tropics that MLS cannot perform. Also, MLS is on-board Aura satellite, which is already well beyond its mission's lifetime, whereas IASI and its successors will be flying for the next decades. We demonstrated in this and previous studies (Emili et al., 2014; Peiro et al., 2018) that the family of IASI sensors is valuable for data assimilation of tropospheric and lower stratosphere $O_3$. This study provides further elements that we believe are important before implementing IASI $O_3$ assimilation in operational systems. We updated the conclusions at page 19 lines 20-22 to remind the importance of assimilating MLS in the stratosphere.

[Figure]

**Figure 3.** SOFRID O$_3$ averaging kernels for the month of July 2010 averaged globally (first plot) and for five latitude bands separately (90°S-60°S, 60°S-30°S, 30°S-30°N, 30°N-60°N, 60°N-90°N). Each coloured line corresponds to a retrieval's level, the corresponding pressure is indicated in the colorbar. Only SOFRID levels with a pressure > 50 hPa are displayed for better clarity.

**2 Reply to specific comments**

1. *Title: "for ozone reanalyses": upon first glance this seems to suggest that the paper presents results of a multi-year reanalysis, which is not the case. Is it necessary to include the word "reanalysis" in the title?*

    **Answer**:

[Figure]

**Figure 4.** Relative difference of RMSE with respect to radiosoundings for MLS-a (teal), MLS+L1a (dark blue) and MLS+L2a (red). This figure replaced Fig. 6 of the original manuscript.

We used the word "reanalyses" because we presented only DA analyses in this study (instead of forecasts). However, we recognize that the study is mostly methodological and does not present results from long simulations. Therefore, we changed the title to "Comparison between the assimilation of IASI Level 2 ozone retrievals and Level 1 radiances in a chemical transport model" to avoid wrong expectations.

5   2. *Abstract, l9: "significant differences". The abstract does not give a very firm conclusion. Does the work presented justify the stronger statement that the non-linearity in the retrievals in combination with unrealistic a-priori profiles are the cause of the L1-L2 differences?*

**Answer**:

Although the results suggest that this seem the case we prefer not to give such a stronger statement in the abstract (and
10   conclusions). The reason is that we did not evaluate explicitly the linearization error of the RTM in our study. Since

[Figure]

**Figure 5.** Taylor diagrams of modeled tropospheric ozone columns (340-750 hPa) for the Control simulation (green), MLS-a (violet), MLS+L1a (grey) and MLS+L2a (yellow) averaged globally and for five latitude bands separately. The Taylor statistics are computed against radiosoundings. This figure replaced Fig. 7 of the original manuscript.

RTTOV is already based on the linearization of a full line-by-line RTM (Saunders et al., 2018), doing this evaluation properly would require implementing the original RTM used by RTTOV in the CTM, which was out of the scope of this study. The main objective of our study was instead to provide some practical answers that can guide future developments for IASI assimilation.

5    3. *Abstract: Is there a clear recommendation from this work? Would L1 assimilation be preferred? A more clear statement would be helpful.*

   **Answer**:

   Our results indicate a slightly better variability of the tropospheric $O_3$ column when assimilating L1 data (Fig. 5). We included this element in the abstract. We also gave different arguments that promote L1 assimilation in the introduction

and in the conclusions (see also Reply n 1), but those are not a direct outcome of our simulations and we prefer addressing the reader to the conclusions to avoid a too lengthy abstract.

4. *Page 2, l5: Useful to mention the averaging kernels as well: .. and DOF linked to the averaging kernels ...*

   **Answer**:

   The sentence has been changed according to the suggestion of the reviewer.

5. *Page 2, l9: "First atmospheric composition models": Reformulate*

   **Answer**:

   The sentence has been reformulated.

6. *Page 2, l19: "However, some aspects of the Data Assimilation (DA) approach differ between the chemistry and meteorology communities." Please be more specific, or remove the sentence.*

   **Answer**:

   The sentence was removed.

7. *Page 2, l23: "This resulted necessary to avoid"; please reformulate.*

   **Answer**:

   The sentence has been reformulated.

8. *Page 2, l24: About the historical background: I was wondering about the problems encountered when assimilating L2 retrievals in NWP (paper of Eyre)? Is it the non-linearity and a-priori dependence (as suggested by the text), or is it a simplification of the retrieval results? The latter could arise when e.g. kernels and full covariance matrices are not used in the assimilation, or not provided by the retrieval teams, which would clearly lead to strong a-priori dependence of the analyses.*

   **Answer**:

   No mention to the averaging kernels was found in Eyre et al. (1993), which suggests that their difficulties arise from the simplification of the retrievals (missing use of averaging kernels). The sentence has been updated to make this point clearer (page 2, line 29).

9. *Page 2, l27: I would suggest to refer to the book of Rodgers as well. Also the paper of Migliorini 2012 is relevant here.*

   **Answer**:

   The references have been updated.

10. *Page 3, l29: "Both are based" It is not so clear what "both" refers to: the two studies, or SOFRID and MOCAGE.*

    **Answer**:

    We refer to SOFRID and MOCAGE DA system, the sentence has been corrected.

11. *Page 4, Section 2.1: METOP also has the GOME2 instrument. Has the synergy IASI-GOME2 been considered? Why the choise to use MLS?*

    **Answer**:

    MLS retrievals are very accurate and provide vertically resolved information that are inaccessible to UV sounders like GOME2. We used MLS to ensure an accurate stratospheric profile and evaluate the impact on IASI TIR assimilation, which was the focus of the study. The assimilation of GOME2 L2 profiles requires some particular care for correcting observation biases (Van Peet et al., 2018) and was not considered for this study. However, we agree with the reviewer's about the interest of performing IASI and GOME2 joint assimilation in future. We included this perspective in the conclusions of the revised manuscript (page 20, lines 1-2).

12. *Page 5, l4: "increased biases": what does "increased" refer to?*

    **Answer**:

    The word increased was removed.

13. *Page 5, l10: "LA" ?*

    **Answer**:

    LA standed for Laboratoire d'Aérologie, it has been replaced by B. Barret.

14. *Page 7, l11: "ECMWF NWP model" Please replace the word "model" by e.g. "NWP model and assimilation system".*

    **Answer**:

    Done

15. *Page 7, l19: The RTTOV versions for the L1 and L2 experiments are different, see table 1. Can the authors be sure that this does not significantly influence the results/conclusions?*

    **Answer**:

    The RTTOV version is now the same for both L1a and L2a and the revised manuscript has been updated based on new results. See also reply n 1 to the first reviewer.

16. *Page 7, l29: "... and was extended ..."*

    **Answer**:

    Correction included.

17. *Page 8, l19: "we assimilate here directly the full L2 profiles (43 levels)". Migliorini wrote a paper (2007) to discuss an efficient interface between L2 retrievals and data assimilation which is relevant in this context. Because the DOF is quite low, this impllies that a lot of noise (43-DOF) is presented to the assimilation when all 43 levels are included. In principle*

*I agree that this avoids any loss of information, but in practice I wonder if the full information may introduce numerical issues (randomness) in the system, especially when this is combined with vertical interpolations? Please comment.*

**Answer**:

We agree with the reviewer about the pertinence of assimilating transformed SOFRID retrievals instead of the full profiles. This would reduce the cost of the L2 assimilation. However, we remind that in this study we used the full L2 error covariance matrices for the assimilation. Hence, the intrinsic noise of each observation level is somehow dampened within the computation of the cost function and its gradient. Source of randomness could result from inaccurate inversion of the observations error covariance matrices. As a matter of precaution, retrievals with inaccurate inverse were already excluded from the assimilation, but they represented less than 0.5% of all the available retrievals (before cloud filtering). Finally, the minimization always showed expected convergence behavior and we did not experience any particular randomness that could be related to numerical issues: repeating the same simulation twice gave same results within the precision of the output format (32 bit floating point). We added a sentence and the appropriate references in the revised manuscript to mention the possibility of using transformed retrievals (page 10, lines 8-11).

18. *Page 8, l22: "The steps for the computation of modeled radiances are equal to the profiles ones until the vertical interpolation." Please reformulate.*

**Answer**:

The sentence has been reformulated.

19. *Page 8, l27: "climatological profile" -> "climatological profiles"*

**Answer**:

Done.

20. *Page 8, l29: "as it is done within SOFRID retrieval scheme" please reformulate*

**Answer**:

Done.

21. *Page 8, l29: Does this mean that the SST is treated differently in the L1 vs L2 assimilation experiments?*

**Answer**:

No. Since SOFRID is a 1D-Var retrieval it does not propagate information to further retrievals as well, it also does not include any SST spatial error covariance. We better underlined the similarities between the two approaches in the revised text (page 10, lines 24-25).

22. *Page 9, l8: "initialized on 1st June 2010"; replace by "initialized on 1 June 2010"*

**Answer**:

Done.

23. *Page 9, l11: "a diagonal matrix (i.e. with no inter-channel correlation) is used". Is the same diagonal matrix used in the retrievals that produce the L2 dataset?*

   **Answer**:

   Yes. It is specified in the sentence before.

24. *Page 9, bottom to p10, top: I got a bit lost with the numbers provided for the background standard deviation, also in comparison with Fig. 1. I understand that a background standard deviation during assimilation is often smaller than the std of a free model run, but I do not manage to connect the numbers with e.g. Fig.1 in combination with Fig.3?! What is the motivation to go from 5% to 2% in the stratosphere, which seems like a big step and does not seem justified given Fig.1? Does this choice lead to very small stratospheric increments? What is the justification for a step between stratosphere and troposphere? The standard deviation should depend on the data assimilated. Normally these kind of numbers are optimised with e.g. a chi-square test.*

   **Answer**:

   The intent of Fig. 1 in the original manuscript was not to provide quantitatively values for the background standard deviation but: i) to display the main features of the error profiles ii) provide the reference values for the following figures that compare different experiments (Fig. 3 and 6 of the original manuscript). The empirical choice of values reported at page 9 resulted from a large number of experiments and we address the reviewer to the reply n 4 to the 1st reviewer for additional arguments that support the standard deviation values used in the end. In the reply n 5 we also show some examples of stratospheric increments that remain significant in terms of $O_3$ concentration.

   The justification of using a step between the stratosphere and the troposphere follows from the vertical features of the errors. The modeled profile is generally more accurate in the stratosphere, especially when MLS is also assimilated (see Fig. 12 of the replies to the 1st reviewer). We are aware of the fact that the background standard deviation depends on the assimilated observations. However, the objective of the study being to compare L1 and L2 assimilation, we did not want to introduce additional differences between the experiments due to different background error covariances. Hence, the most pragmatic option was to find a sort of compromise that fits reasonably well for all the presented experiments. The chi-square test is a useful diagnostics in DA but we did not consider it appropriate in our study for the following reasons: i) we use a very simplified observation error matrix and optimizing only the background error but keeping R fixed does not seem relevant ii) it is generally not possible to keep an optimum chi-square when using the same B but changing the assimilated instruments.

   We included elements from these replies in the revised manuscript (page 12 and top of page 13) to better explain the reasoning behind the choices for B.

25. *Page 10, l15: One would expect that features in the boundary layer, and, to a lesser extent, the free troposphere show vertical correlations because of e.g. vertical mixing and convection. This in contrast to the stratosphere.*

   **Answer**:

We agree in principle with the reviewer, and we did test a configuration with larger vertical correlations in the troposphere (1.5 model levels) than in the stratosphere (0.5 model levels), but results were not significantly better. We updated the text to include this element (page 12, lines 23-25, page 13, lines 1-2).

26. *Page 10, first part: I think the B matrix discussion can be shortened somewhat, because optimising it is not so important for the topic of the paper.*

    **Answer**:

    We removed some sentences corresponding to the settings employed in previous studies since they were not strictly necessary.

27. *Page 11, l5: "are in generally ", remove "in"*

    **Answer**:

    Done.

28. *Page 11, l7: "are found in correspondence of tropical latitudes". Please reformulate.*

    **Answer**:

    Done.

29. *Page 11, l14: "equivalence between L1 and L2 assimilation is not verified for O3". I suggest to explicitly add "for O3 retrievals in the thermal infrared".*

    **Answer**:

    Done.

30. *Page 12, l2: "The assimilation increases the RMSE of the tropospheric profile at northern latitudes (60N-90N)." I guess you mean in the range 350-1000 hPa.*

    **Answer**:

    The range has been added in parenthesis.

31. *Page 12, l8: "the other way round". Replace by "around".*

    **Answer**:

    Done.

32. *Page 12, l17: "Hence, we expect a stronger impact of the prior in the retrieval results,". I do not understand this. It means that the DOF is smaller, which is clearly observed at altitudes around 200hPa, where the improvement with IASI data is much more limited in the SH. But a-priori plays only a role through non-linear effects. Why would these non-linear effects be larger in the SH?*

**Answer**:

The reviewer's comment was very pertinent: our conclusion was not supported by the data (Fig. 1) but was a wrong interpretation of the original results (see reply n 1 to the 1st reviewer): the new experiments show indeed similar results also in the SH mid-latitudes. The discussion has been revised according to the new results.

33. *Page 12, l27: "The only exceptions are a lower RMSE degradation at 50 hPa". Should we believe the sondes or MLS here? How many sonde launches are included, and confirm the 60 hPa bias?*

**Answer**:

The number of radiosoundings has been reported in the revised manuscript, and we believe that higher confidence should be given to the MLS validation, especially with respect to standard deviation values. This aspect has been better highlighted in the revised manuscript (page 12, line 10 and page 16, lines 10-12).

34. *Page 12, l32: "total computing time is 3.9 CPU hours". Is this on a single core/node ??*

**Answer**:

The simulations have been performed on one Xeon E5-2680 node with 24 cores. The values given in hours are expressed in CPU time, which depends on the CPU type / frequency and give an approximate idea of the relative computational cost of L1a versus L2a. The run time (or elapsed time), given in minutes, depends on the parallel implementation and the number of cores/nodes used for the simulations. We replaced "total computing time" by "total CPU time" to avoid confusion.

35. *Page 13, Fig.5: The figure seems to prove that the analysis of MLS and of IASI are more consistent in the case of L1, while the L2 plot indicates biases between the instruments, especially in the tropics. This could be discussed a bit more explicitly.*

**Answer**:

Following the comment n 6 of the 1st reviewer, we replaced Fig. 5 with the MLS+L1a - MLS+L2a averages. The discussion has been revised accordingly.

36. *Page 14, l16: "mixed elsewhere"? Do you mean to say "mixed results are obtained elsewhere"?*

**Answer**:

The sentence has been changed.

37. *Page 14, l22: "which are differences" please correct the English.*

**Answer**:

The phrase has been corrected.

38. *Page 14, l23: "reanalyses". I suggest to broaden this to e.g. "analyses and reanalyses".*

    **Answer**:

    Done.

39. *Page 14, l24: "between the L2 retrieval and the assimilation algorithm ". I suggest to change to " between the L2 retrieval and the L1 observation operator" or something like that*

    **Answer**:

    Since we also extended the control vector of the assimilation, we rephrased with: "between the L2 retrieval and the L1 assimilation".

40. *Page 14, l24: "using the same RTM". But the version of the RTM is different ?!*

    **Answer**:

    The version of the RTM is now the same in the revised manuscript.

41. *Page 14, l28: "between each other". I suggest "against each other".*

    **Answer**:

    Corrected.

42. *Page 14, l30: "Main findings suggest". I suggest "The results suggest ..."*

    **Answer**:

    Corrected.

43. *Page 15, l6: "We could imagine". I could as well, but is this a recommendation?*

44. *Page 15, l6: The non-linearity of the retrieval may be very different for different species and spectral ranges. Which ones would be candidates to show significant differneces between L1 and L2?*

45. *Page 15, l9: I was wondering if we may expect positive synergies between IASI and GOME2? They are both on the same platform. Please discuss.*

    **Answer**:

    This paragraph groups the the answers to the above 3 comments.

    The original text at page 15, lines 3-8 was replaced since it was not appropriate anymore with the new results (reply n 1 to the first reviewer). We don't have enough experience with other retrievals than $O_3$ to give detailed recommendations on which species and L2 product might be affected by similar issues. Preliminary analyses indicate that $O_3$ profile retrieval in the UV region might a display similar behavior than in the TIR. However, the degree of non-linearity depends significantly on the retrieval's a-priori: a case-by-case analysis would be needed in this sense. We included the perspective

of GOME2 assimilation in the revised manuscript and recommended to analyze potential dependence of the results to the a-priori (page 19, line 23).

46. *Page 15, l15: "Level 2 products can be aggregated". This useful remark could be phrased more generally by refering to "Use of the Information Content in Satellite Measurements for an Efficient Interface to Data Assimilation" by Migliorini et al, 2007. Through L2 the number of useful observations presented to the assimilation may be optimised (to ultimately match the DOF).*

**Answer**:

The methodology of Migliorini et al. (2008) is now referenced and briefly discussed in Sec. 3.3 (reply n 17). At page 15, line 15 we were specifically addressing models that does not cover the full atmosphere and the case of vertical selection/aggregation of measurements based on user needs. To our understanding, the method proposed by Migliorini et al. (2008) does not seem to be adapted this type of needs because it compresses the full retrieval's information.

**References**

Emili, E., Barret, B., Massart, S., Le Flochmoen, E., Piacentini, a., El Amraoui, L., Pannekoucke, O., and Cariolle, D.: Combined assimilation of IASI and MLS observations to constrain tropospheric and stratospheric ozone in a global chemical transport model, Atmospheric Chemistry and Physics, 14, 177–198, https://doi.org/10.5194/acp-14-177-2014, http://www.atmos-chem-phys.net/14/177/2014/, 2014.

5 Eyre, J. R., Kelly, G. A., McNally, A. P., Andersson, E., and Persson, A.: Assimilation of TOVS radiance information through one-dimensional variational analysis, Quarterly Journal of the Royal Meteorological Society, 119, 1427–1463, https://doi.org/10.1002/qj.49711951411, https://rmets.onlinelibrary.wiley.com/doi/abs/10.1002/qj.49711951411, 1993.

Migliorini, S., Piccolo, C., and Rodgers, C. D.: Use of the Information Content in Satellite Measurements for an Efficient Interface to Data Assimilation, Monthly Weather Review, 136, 2633–2650, https://doi.org/10.1175/2007MWR2236.1, http://journals.ametsoc.org/doi/abs/

10 10.1175/2007MWR2236.1, 2008.

Peiro, H., Emili, E., Cariolle, D., Barret, B., and Le Flochmoën, E.: Multi-year assimilation of IASI and MLS ozone retrievals: variability of tropospheric ozone over the tropics in response to ENSO, Atmospheric Chemistry and Physics, 18, 6939–6958, https://doi.org/10.5194/acp-18-6939-2018, https://www.atmos-chem-phys.net/18/6939/2018/, 2018.

Saunders, R., Hocking, J., Turner, E., Rayer, P., Rundle, D., Brunel, P., Vidot, J., Roquet, P., Matricardi, M., Geer, A., Bormann, N., and Lupu,

15 C.: An update on the RTTOV fast radiative transfer model (currently at version 12), Geoscientific Model Development, 11, 2717–2737, https://doi.org/10.5194/gmd-11-2717-2018, https://www.geosci-model-dev.net/11/2717/2018/, 2018.

Van Peet, J. C., Van Der, R. J., Kelder, H. M., and Levelt, P. F.: Simultaneous assimilation of ozone profiles from multiple UV-VIS satellite instruments, Atmospheric Chemistry and Physics, 18, 1685–1704, https://doi.org/10.5194/acp-18-1685-2018, 2018.

Ziemke, J. R., Chandra, S., Labow, G. J., Bhartia, P. K., Froidevaux, L., and Witte, J. C.: A global climatology of tropospheric

20 and stratospheric ozone derived from Aura OMI and MLS measurements, Atmospheric Chemistry and Physics, 11, 9237–9251, https://doi.org/10.5194/acp-11-9237-2011, http://www.atmos-chem-phys.net/11/9237/2011/, 2011.